# Precise allele-specific genome editing by spatiotemporal control of CRISPR-Cas9 via pronuclear transplantation

Yanhe Li[1], Yuteng Weng[2], Dandan Bai[1], Yanping Jia[1], Yingdong Liu[1], Yalin Zhang[1], Xiaochen Kou[1], Yanhong Zhao[1], Jingling Ruan[1], Jiayu Chen [1], Jiqing Yin[1], Hong Wang[1], Xiaoming Teng[1], Zuolin Wang[2], Wenqiang Liu[1✉] & Shaorong Gao [1✉]

Gene-targeted animal models that are generated by injecting Cas9 and sgRNAs into zygotes are often accompanied by undesired double-strand break (DSB)-induced byproducts and random biallelic targeting due to uncontrollable Cas9 targeting activity. Here, we establish a parental allele-specific gene-targeting (Past-CRISPR) method, based on the detailed observation that pronuclear transfer-mediated cytoplasmic dilution can effectively terminate Cas9 activity. We apply this method in embryos to efficiently target the given parental alleles of a gene of interest and observed little genomic mosaicism because of the spatiotemporal control of Cas9 activity. This method allows us to rapidly explore the function of individual parent-of-origin effects and to construct animal models with a single genomic change. More importantly, Past-CRISPR could also be used for therapeutic applications or disease model construction.

---

[1] Clinical and Translational Research Center of Shanghai First Maternity and Infant Hospital, Shanghai Key Laboratory of Signaling and Disease Research, Frontier Science Center for Stem Cell Research, School of Life Sciences and Technology, Tongji University, 200092 Shanghai, China. [2] Shanghai Engineering Research Center of Tooth Restoration and Regeneration, Department of Implantology, School and Hospital of Stomatology, Tongji University, 200092 Shanghai, China. ✉email: liuwenqiang@tongji.edu.cn; gaoshaorong@tongji.edu.cn

CRISPR-Cas9 can be programmed with single-guide RNAs (sgRNAs) that recognize specific genomic sequences and induce double-strand breaks (DSBs) in vitro and in vivo[1–5]. Then, nonhomologous end-joining (NHEJ) mechanisms are primarily activated and are preferentially used to introduce additional mutations at the break sites, resulting in frameshift mutations[6]. Given the relatively high efficiency of NHEJ, co-injection of Cas9 mRNA and sgRNAs into pronuclear stage zygotes has become a powerful gene-targeting tool to generate genome modification in various species[7,8]. Nonetheless, the constitutively active nature of Cas9 makes it difficult to achieve precise spatial and temporal control over Cas9 gene-editing activity, which leads to numerous limitations and unpredictable hazards[7,9,10]. First, the majority of generated gene-edited embryos showed mosaic genotypes in different blastomeres due to prolonged Cas9 on-targeting activity after zygotic division[11]. Second, monoallelic site editing is ineffective under the traditional CRISPR-Cas9 system because the cleavage efficiency of the CRISPR-Cas9 system is spatiotemporally uncontrollable[12–14]. Thus, parental allele-specific genomic sites cannot also be selectively recognized and modified by the traditional CRISPR-Cas9 system. These defects reveal that a lack of control over Cas9 activity limits the rapid and economical properties of CRISPR-Cas9 for research on gene function and for clinical application[12,15,16]. Numerous strategies have been developed to control Cas9 activity via modifying the Cas9 protein or its programming gRNA molecules[12–14,16], but such strategies are still at an exploratory stage.

In this study, we develop a method that allows spatiotemporal control of Cas9 activity by pronuclear transplantation. Diluting the amounts of Cas9:sgRNA in vivo terminates its enzymatic activity following on-target editing. This spatially improves allele selectively to achieve highly efficient parental allele-specific gene editing, and it temporally restricts Cas9 activity to a narrow window, which can dramatically reduce mosaicism.

## Results

**Overviews of Past-CRISPR**. To date, there have been no satisfactory methods for controlling the targeting activity of Cas9 injected into embryos. We assume that the amounts of Cas9 mRNA and sgRNAs can be diluted through degradation following pronuclear transplantation into zygotes, which might achieve spatiotemporally controlled Cas9 activity. In addition, the zygotic pronucleus stage is the unique window where the parental genomes are physically separated across the life cycle, probably giving us a space to achieve parental allele-specific genome targeting (Past-CRISPR).

To test our hypothesis, we selected *Anapc2* as an example gene to assess the feasibility of this method. *Anapc2* has been shown to be necessary for preimplantation embryo development[17]. Embryos with homozygous deletion of *Anapc2* ($Anapc2^{-/-}$) were arrested at the morula stage. Past-CRISPR was performed by co-injecting Cas9 mRNA and sgRNAs into MII stage oocytes and then performing in vitro fertilization (IVF). After 7–9 h of IVF, we isolated paternal or maternal pronucleus from pronucleus stage 3–4 (PN3–4) zygote and transferred them into wild-type embryos that had been depleted of paternal or maternal pronucleus, respectively. The pronucleus was fused with the cytoplasm by the Sendai virus and generated reconstructed normal diploid zygotes; theoretically one of the two sets of DNA had been edited, and the other was wild type (Pat-ko/Mat-wt or Mat-ko/Pat-wt) (Fig. 1a, Supplementary Fig. 1a). The traditional method of zygote microinjection of Cas9 mRNA and sgRNAs was used as a control group for comparison with Past-CRISPR.

**Allele-specific gene targeting achieved by Past-CRISPR**. Immunofluorescence staining analysis was performed for Cas9 components in the zygotes and two-cell embryos generated by the traditional zygote injection method and Past-CRISPR targeting *Anapc2*, and the results showed that the zygote reconstructed by Past-CRISPR displayed a weak Cas9 signal in the nuclear region (Fig. 1b). Instead, Cas9 showed obvious staining throughout the nucleoplasm in the contemporaneous zygote that underwent zygote injection and was still detectable at the two-cell stage. These results revealed that residual Cas9 components remained during development to the two-cell stage following zygote injection, whereas pronuclei transferred from Cas9-injected zygotes into enucleated wild-type embryos via Past-CRISPR contained dramatically diluted Cas9:sgRNA that were undetectable by immunofluorescence staining. Notably, unlike *Anapc2* knockout embryos arrested at the morula stage, we found that the *Anapc2* paternal allele-specific edited embryos and the *Anapc2* maternal allele-specific edited embryos generated by Past-CRISPR both can develop to the blastocyst stage, which is the same as the control groups of wild-type embryos and mat/pat-swapped embryos (Fig. 1c, d, Supplementary Fig. 1b, c). We randomly assessed these three groups of edited embryos (*Anapc2*-targeted embryos, *Anapc2* mat-edited and pat-edited embryos) by PCR amplification and TA clone sequence analysis (see Methods for details). Similar to previous reports in mice, mutations were induced by the traditional methods of co-injection of Cas9 mRNA and sgRNA into zygote and all embryos carried biallelic targeting with no wild-type alleles. However, 13 of the 17 embryos carried more than two mutant genotypes (average 3.58 types of mutation alleles among zygote injected embryos), and many alleles carried indels without forming frameshift mutations (Supplementary Fig. 1d and Supplementary Table 1). We further performed an accurate analysis of the gene-edited four-cell embryos at the single-cell level (Supplementary Fig. 1e, f and Supplementary Table 2). The results showed that almost every blastomere carried a mutant genotype that was different from each of the others in the *Anapc2*-targeted embryos, among which alleles did not form frameshift mutations in many blastomeres. These results suggest that most embryos displayed genetic mosaicism. Furthermore, we found that these embryos all carry biallelic mutations, and no monoallelic mutations were detected, which was consistent with previous studies[18]. These data confirmed the high efficiency of generating biallelic mutations in vivo by the traditional zygote injection methods; however, there was difficulty in producing monoallelic mutations, and the continuous nuclease activities provided uncontrollable generation of additional indels in early cleavage-stage embryos that caused the formation of mosaic embryos. Remarkably, the TA cloning outcomes for each embryo edited by Past-CRISPR show only uniform indel alleles and wild-type alleles (Fig. 1e and Supplementary Fig. 2a). Moreover, among all the alleles, 52.9% (36/68) of these clones carried indels, and the remainder (47.1%, 32/68) were wild-type (Fig. 1f). We then use TIDE (Tracking Indels by Decomposition, Desktop Genetics)[19] analysis as a sequence trace decomposition tool to further prove the accuracy of the above data (see Methods for details). Similar results were presented in the sequence traces outcomes (Supplementary Fig. 2b, c). The results together indicated that Cas9-induced DSB formation occurs just before the first round of DNA replication in one-cell embryos and that the *Anapc2* potential parental allele-specific edited embryos were uniformly modified on one *Anapc2* allele by Past-CRISPR.

To more accurately assess the parental allele-specific targeting in the embryos, we used C57BL/6J oocytes and PWK sperm to construct hybrid embryos; these two strains have abundant single nucleotide polymorphisms (SNPs) throughout their genomes,

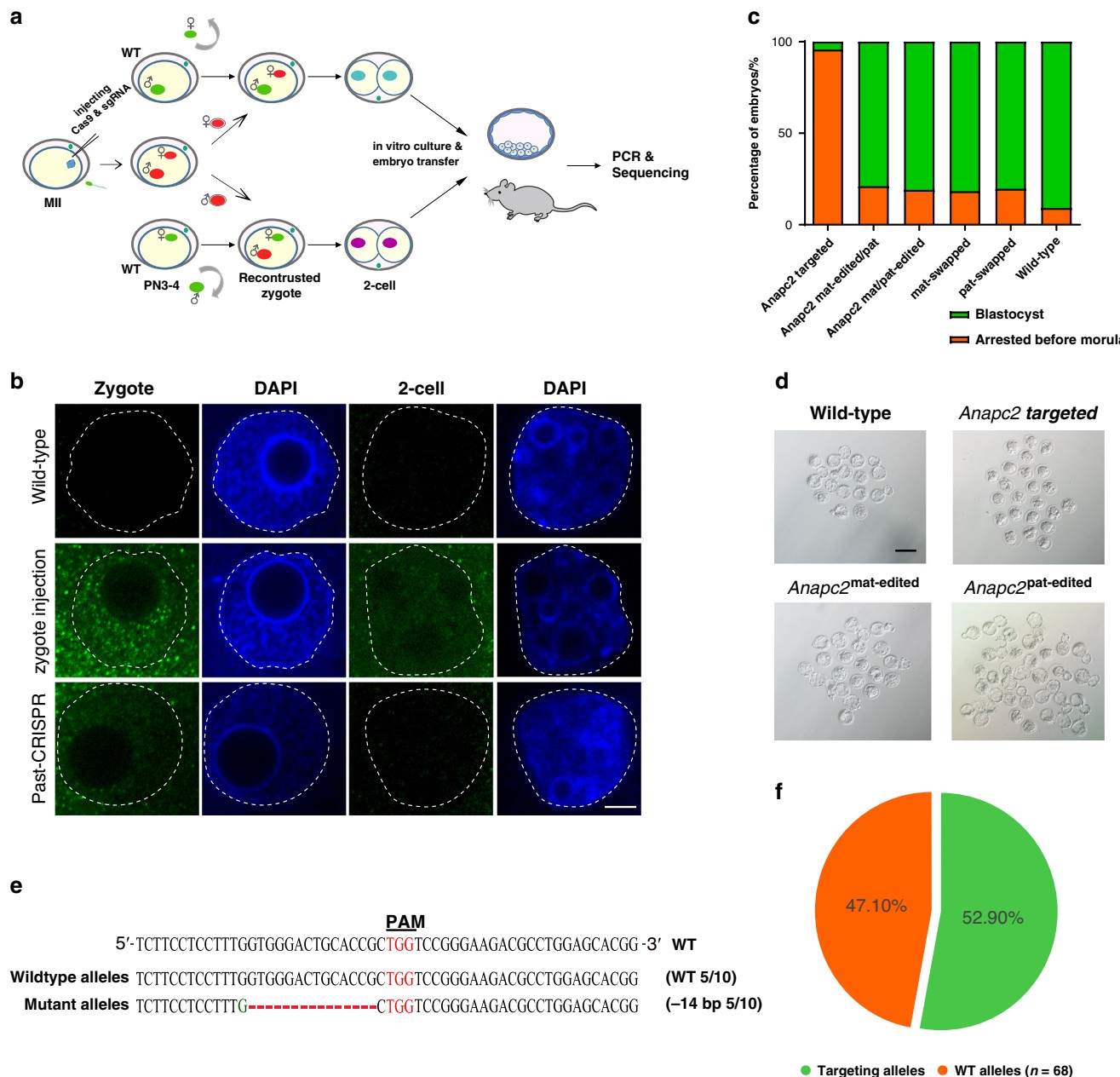

**Fig. 1 The design and validation of parental allele-specific site targeting methods. a** A mixture of Cas9 and an individual sgRNA targeting *Anapc2* was injected into MII oocytes. The injected oocytes and wild-type oocytes underwent IVF simultaneously. When the injected embryos developed into pronuleus PN3–4 stage 7–9 h post IVF, their paternal or maternal pronuclei (red ovals) were isolated and transferred into wild-type embryos with the maternal or paternal pronuclei (green ovals) depleted. The pronuclei were fused with the cytoplasm by the Sendai virus and generated reconstructed normal diploid zygotes. Some of the reconstructed embryos were transplanted into oviducts of pseudopregnant females, and others were cultured in vitro. Embryos and offspring were generated for PCR and sequencing. **b** Immunostaining for detecting SpCas9 signals in the nuclear region of zygotes and 2-cell embryos resulting from Past-CRISPR and traditional zygote injection. White dashed lines demarcate the nuclear membrane. The data shown are representative of at least two independent experiments. Scale bars, 5 µm. **c** A bar graph shows the developmental potential (counted 96 h after fertilization) of *Anapc2*-targeted embryos, *Anapc2*^mat-edited/pat embryos, *Anapc2*^mat/pat-edited embryos, and the wild-type groups (mat/pat-swapped wild-type embryos and wild-type embryos). **d** Bright field images of *Anapc2*-targeted embryos, *Anapc2*^mat-edited/pat embryos, *Anapc2*^mat/pat-edited embryos and wild-type embryos. One representative result of three independent experiments is shown. Scale bars: 100 µm. **e** TA cloning sequences of the *Anapc2* gRNA target sites from one of the potential parental allele-specific edited embryos. Sequences of ten alleles present in individual embryos reveal two different types of alleles: half are wild-type alleles, and the other half are mutant alleles. **f** A pie chart demonstrates proportions of targeting alleles and wild-type alleles in the tested *Anapc2* potential parental allele-specific edited embryos. Number, total alleles counted.

including the gene *Anapc2*, which enables differentiation of the parental alleles (Supplementary Fig. 3a). *Anapc2* potential paternal allele-specific edited blastocysts and *Anapc2* potential maternal allele-specific edited blastocysts (*n* = 15) obtained by Past-CRISPR were assessed by PCR amplification and TA clone sequencing, and the results showed that 87% (13/15) of these blastocysts had monoallelic *Anapc2* depletion (Supplementary Table 3). More importantly, all the parental allele-specific edited sites in these monoallelic depleted embryos uniformly matched the respective parental SNP sites (Fig. 2a and Supplementary

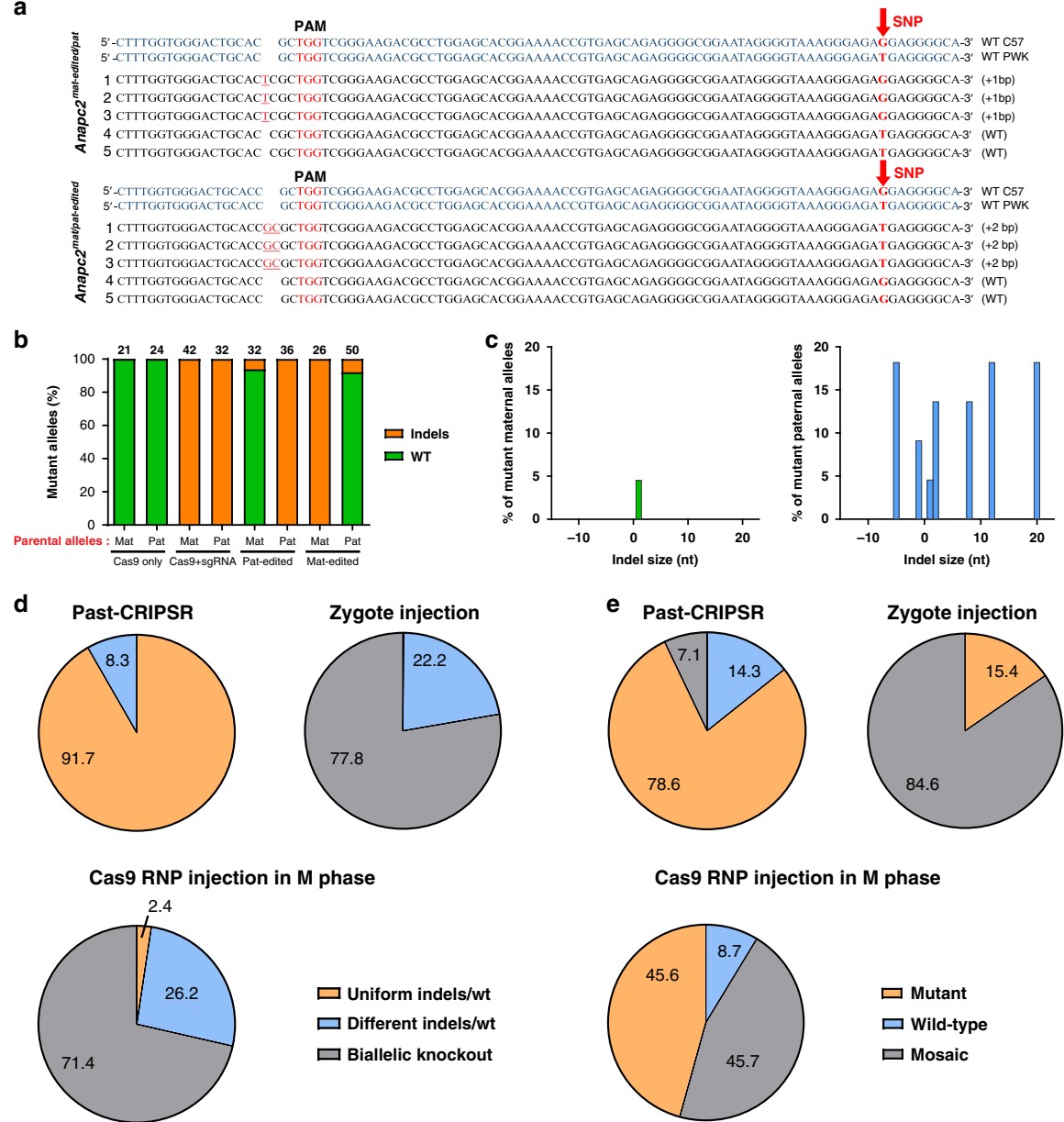

**Fig. 2 Functional assessments for Past-CRISPR methods. a** Sequence of alleles present in individual *Anapc2*mat-edited/pat and *Anapc2*mat/pat-edited hybrid embryos constructed from C57BL/6J oocytes and PWK sperm. The SNP site between C57BL/6J and PWK is indicated by red arrows. Number, a total of 15 embryos counted. **b** On-target allele analysis of TA clone sequencing of blastocysts treated with Cas9-only, Cas9+sgRNA, Pat-edited, and Mat-edited. The number above each column represents the total clones analyzed. **c** Indel profiles from embryos who were performed paternal allele edited by Past-CRISPR. *Anapc2* maternal alleles (left) and paternal alleles (right) are plotted separately. Minus numbers represent deletions, and plus numbers represent insertions. Statistics for sequences without indels and with single-base substitution mutation indels are not shown in this chart. **d** Pie charts show the percentage of the three predicted targeting genotypes in the Past-CRISPR- edited embryos (upleft), traditional zygote injection embryos (upright) and Cas9 RNP injection in M-phase embryos (downleft). Numbers in pie charts represent the percentage of embryos. All the tested embryos were from at least six independent experiments. **e** Pie charts showing percentages of different genotypes of all embryos treated with Past-CRISPR (upleft), traditional zygote injection methods (upright) and Cas9 RNP injection in M-phase (downleft). Numbers in pie charts represent the percentage of embryos. All the tested embryos were from at least three independent experiments.

Fig. 3b). Allele-specific analysis revealed that 92–94% of all indels that occurred in the parental allele-specific edited embryos were present in the specified parental allele. By contrast, traditional zygote injection methods could not selectively target the specified allele (Fig. 2b, Supplementary Fig. 3c). The indel profile showed that most of the Cas9-induced variants caused a frameshift. There were few indels (less than 7%) present in the incorrect parental allele (Fig. 2c, Supplementary Fig. 3d). These results suggest that parental allele-specific sites could be selectively recognized and

modified by Cas9 with a one-step process with the help of Past-CRISPR.

**High monoallelic targeting efficiency of Past-CRISPR method.** To test the universality and monoallelic targeting efficiency of Past-CRISPR, we continued to analyze the targeting outcomes by enlarging the sample size of parental allele-specific edited embryos (*n* = 70) and analyzed different genes (*Anapc2*, *Peg10*,

Mash2), and we confirmed that the 85.7% (60/70) targeting efficiency was comparable to the 100% (26/26) efficiency of traditional zygote injection methods (Supplementary Fig. 4a and Supplementary Table 4). Only three targeting genotypes could be detected in all targeted outcomes (uniform indels/wt, different indels/wt, double indels, Supplementary Fig. 4b). Sequencing of the 60 gene-targeting embryos by Past-CRISPR revealed that 55 (91.7%) were monoallelic mutants without a mosaic genotype (uniform indels/wt). Only 5 (8.3%) embryos were monoallelic mutants with a mosaic genotype (different indels/wt), and none of the embryos were double knockouts (uniform or mosaic homozygous knockout) (Fig. 2d). The efficiency of monoallelic mutant production for the three genes and between pat-edited embryos and mat-edited embryos was approximately similar (Supplementary Fig. 4c, d), and the ratio between targeted alleles and wild-type alleles was even closer in the large sample size (Supplementary Fig. 4e), suggesting that this method has universal and highly efficient effects on targeting different genes. Meanwhile, we found that among the 70 embryos, only 7.1% (5/70 embryos carried >1 type of indels) had genomic mosaicism, while traditional zygote injection methods resulted in more than 84.6% (22/26 embryos carried >2 types of indels) of gene-edited embryos having mosaicism (Fig. 2e, Supplementary Table 4). The injection of MII oocytes with Cas9 RNPs (Cas9:sgRNA ribonucleotide protein complexes) at fertilization has been reported to lead to lower mosaicism[20,21], so we also examined the same gene-editing strategy by targeting Anapc2. However, despite obvious reduced mosaic efficiency, the sequencing results still lead to at least 45.7% (21/46 embryos carried >2 types of indels) of the gene targeting embryos having mosaicism, and only one of 42 (2.38%) of the gene-targeting embryos was a monoallelic mutant (uniform indels/wt, Fig. 2d, e). Overall, these results support the hypothesis that these parental allele-specific edited embryos exhibited significantly reduced mosaicism and high monoallelic targeting efficiency compared with that of the zygote injected embryos and Cas9 RNP injection in M-phase embryos.

**The generation of embryonically lethal mutation animals.** The construction of viable animal models with heritable embryonically lethal mutations still remains elusive[7,18]. Usually, none of the pups were born by injecting Cas9 mRNA and sgRNA targeting embryonically lethal genes into zygotes, which prevents the generation of viable mouse models with a heritable embryonically lethal mutation. Here, we tested whether Past-CRISPR could directly generate Anapc2 monoallelic knockout mice. Sixty maternal allele edited embryos were transplanted into the oviducts of pseudopregnant females, and eight mice were born, of which 50% were monoallelic mutants carrying frameshift mutations that survived for more than 4 months (Fig. 3a, Supplementary Fig. 5a and Supplementary Table 4). To test the transmissibility of lethal gene mutations, we crossed the F0 heterozygous mice ($Anapc2^{+/-} \times Anapc2^{+/-}$), and 78% F1 mice were born with monoallelic mutations; no mice were born with biallelic mutations (Fig. 3b, c), which further confirmed that Anapc2 is an embryonically lethal gene. The results suggested that Past-CRISPR could offer a rapid and efficient process to generate monoallelic mutant mice and generate animal models carrying heritable lethal mutations further for studying lethal gene function (Supplementary Fig. 5b).

**Functional verification of individual imprinted genes.** Next, we explored Past-CRISPR for imprinted gene functional studies in vivo. Genomic imprinting is responsible for parental allele-specific regulation of gene expression, which is essential for mammalian development[22]. Importantly, recent studies have also reported many noncanonical imprinted genes established and maintained during the preimplantation stage, most likely accounting for embryonic development[23,24]. The traditional verification procedure is currently costly and time consuming, and it requires multiple steps and further serial crossbreeding[25–28]. To address this question, we chose Mash2 and Peg10, two genes that are parental allele-specific expressed imprinted genes[26,27]; the early embryonic lethal phenotype could be confirmed by the traditional zygote injection methods directing the depletion of each gene, which is consistent with the gene disruption mutants by homologous recombination reported previously[26,27] (Supplementary Fig. 6). We attempted to generate Mash2/Peg10^mat-wt/pat-ko and Mash2/Peg10^mat-ko/pat-wt embryos in the F0 generation by Past-CRISPR (Supplementary Fig. 7a, b and Supplementary Table 4). We found that the development of embryos between the groups was almost normal in the preimplantation stage compared to that of the wild-type embryos. However, Mash2^mat-ko/pat-wt ($p = 0.0007$ vs wild type, df (2, 26), $F = 10.44$, one-way ANOVA) and Peg10^mat-wt/pat-ko embryos ($p < 0.0001$ vs wild type, df (2, 32), $F = 10.44$, one-way ANOVA) generated in F0 by Past-CRISPR both exhibited severe growth retardation after 10.0 d.p.c. (Fig. 3d, e). This phenotype is consistent with previously published work[26,27]. The outcomes of these targeting experiments were that the embryos were monoallelic mutants (Supplementary Fig. 7c). Furthermore, the generated Mash2^mat-wt/pat-ko embryos ($p = 0.898$ vs wild type, df (2, 26), $F = 15.15$, one-way ANOVA) and Peg10^mat-ko/pat-wt embryos ($p = 0.999$ vs wild type, df (2, 32), $F = 15.15$, one-way ANOVA) are similar to the wild-type embryos, which could survive through 10.0 d.p.c. (Fig. 3d, e). By Mash2 paternal allele editing, seven pups were generated, among which five of the genotyped pups were Mash2^mat-wt/pat-ko. By Peg10 maternal allele editing, nine pups were generated, among which seven of the genotyped pups were Peg10^mat-ko/pat-wt (Supplementary Fig. 7c–e). Thus, compared to the multistep process of traditional verification, the Mash2 and Peg10 parental allele-specific embryos could be generated in F0, and the phenotype verification could be completed with a one-step process by Past-CRISPR (Supplementary Fig. 7f). This allows for a relatively easy in vivo study of the parent-of-origin effects, including functions of candidate imprinted genes.

**The generation of identical biallelic gene knockout animals.** Since single pronucleus isolated and fused with the cytoplasm of a wild-type embryo by Past-CRISPR could contribute to identical indels on a single allele of a diploid embryo, we explored whether biallelic identical indels could be generated at the genome of every cell and achieved complete gene knockout without a mosaic genotype. Therefore, we performed an extreme strategy of performing two-pronucleus (2PN) transfer at the PN3–4 stage based on Past-CRISPR and targeted the Tyrosinase gene ('Tyr' for short) to determine the type of mutation[15,29] (Fig. 3f), as mice with homozygous loss-of-function show an obvious albino phenotype and incomplete mutants show mosaicism of pigmentation. By comparing the three gene-editing strategies, we found that 57.1% of albino mice and 42.9% of spotted mice were generated by traditional zygote injection, and 20% of albino mice and 30% of spotted mice are generated with Cas9 RNP injection in M-phase. However, although 40% of mice generated by 2PN transfer strategies were normal, only 10% of mice showed slight spotted pigmentation (Fig. 3g, h). DNA genotyping analysis results show that embryos generated by 2PN transfer strategies show lower mosaicism (16.6%) than embryos generated by zygote injection (81.8%) and embryos generated by Cas9 RNP injection in M-phase (44.44%) (Fig. 3i). Meanwhile, the editing efficiency (50%) for biallelic mutations with identical indels by 2PN transfer strategies is 2.8-fold respectively compared to zygote injection

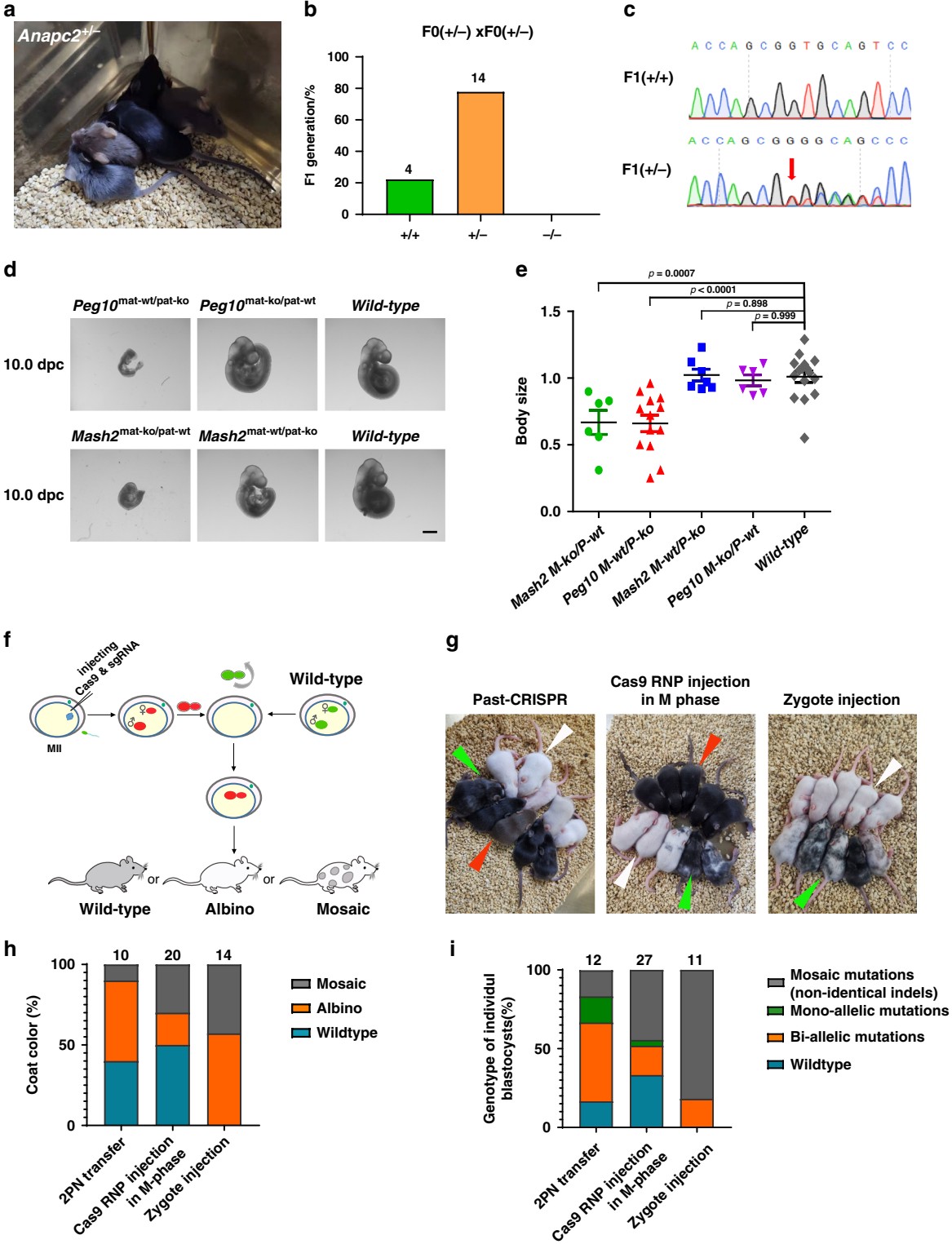

(18.2%) and Cas9 RNP injection in M-phase (18.53%). Taken together, these results suggest that the 2PN transfer strategy based on the Past-CRISPR achieved targeted biallelic mutations with identical indels more efficiently than what was achieved by traditional zygote injection or Cas9 RNP injection in M-phase strategy. Thus, we believe that these strategies would be more valuable than traditional strategies for application in the production of gene knockouts in large animals and of difficult to breed animals without producing genotypic mosaicism.

**Specific disrupting of dominant alleles to block dwarfism.** Next, we want to test whether Past-CRISPR can be used for treating dominant genetic diseases. By analyzing the human genetic variants in the ClinVar database[30], we screened 1789 dominant pathogenic variants from a total of 81,137 pathogenic entries (Supplementary Fig. 8a and Supplementary Data 1, 2), from which the function of the dominant mutant allele affects the phenotype even in the presence of a wild-type allele. Thus, allele-specific disruption of the dominant allelic mutations could remove

**Fig. 3 Application to quickly determine genomic imprinting and construction of animal models. a** Representative image shows heterozygous *Anapc2* mutant mice generated by Past-CRISPR and have survived for more than 4 months. **b** Reproductive ability and germline transmission of *Anapc2*$^{+/-}$ mice resulting from Past-CRISPR treatment. Two female *Anapc2*$^{+/-}$ mice were crossed with two male *Anapc2*$^{+/-}$ mice individually and two litters of pups were born. Number, total number of mice counted. **c** Sequencing traces of PCR products encompassing the *Anapc2* target region from a representative wt offspring (F1) (upper) and heterozygous mutation offspring (F1) (lower) of *Anapc2*$^{+/-}$ mice generated by Past-CRISPR. The mutation site is marked by a red arrow. **d** Representative images of 10.0 d.p.c *Mash2/Peg10* parental allele-specific knockout embryos and wild-type embryos. One representative result of at least three independent experiments is shown. Scale bars: 1 mm. **e** Body size for wild-type embryos ($n = 16$), *Mash2*$^{mat-ko/pat-wt}$ embryos ($n = 6$), *Peg10*$^{mat-wt/pat-ko}$ embryos ($n = 13$), *Mash2*$^{mat-wt/pat-ko}$ embryos ($n = 7$) and *Peg10*$^{mat-ko/pat-wt}$ embryos ($n = 6$) at 10.0 d.p.c were calculated and are shown. Each dot represents individual embryos with the median indicated by the horizontal lines. Data were expressed as mean ± SEM. *P* values (one-way ANOVA) are also shown. These embryos in each group were counted from at least two individual experiments. Source data are provided as a Source data profile. **f** Experimental design. A mixture of Cas9 and sgRNA targeting *Tyr* was injected into MII oocytes. Injected oocytes and wild-type oocytes were simultaneously subjected to IVF. When the injected embryos developed into PN3–4 stage 8 h post IVF, the two pronuclei were swapped into enucleated wild-type embryos. The reconstructed embryos were transplanted into the oviducts of pseudopregnant females. Offspring were generated for observation. **g** Pigmentation phenotypes of representative litters resulting from 2PN-transfer (left), Cas9 RNP injection in M-phase and traditional zygote injection (right). White arrowhead, albino; red arrowhead, wild type; green arrowhead, spotted. **h** Percentages of different phenotypes of mice resulting from 2PN transfer, Cas9 RNP injection in M-phase and traditional zygote injection. The number above each column represents the total mice counted. **i** Percentages of different mutation types of blastocysts resulting from 2PN-transfer, Cas9 RNP injection in M-phase and traditional zygote injection. The number above each column represents the total number of embryos analyzed.

restrictions on the wild-type alleles and overcome dominant-negative effects, which is likely to be realized by Past-CRISPR (Fig. 4a). Accordingly, we focused on a mouse model with a dominant point mutation in *Fgfr3* (*Fgfr3* Gly369Cys mutation), causing dwarfism with features mimicking human achondroplasia; further, homozygous mutant mice have more severe effects than heterozygotes[31,32]. To explore Past-CRISPR efficiently, we designed two candidate sgRNAs targeting exon 4 of *Fgfr3* in front of exon 8, which harbors the dominant mutation (Supplementary Fig. 8b). SgRNA1 induced monoallelic mutations more efficiently (average 79%) than sgRNA2 (average 4.8%) and with a low frequency of mosaicism (average 10.5%) (Fig. 4b). Thus, we used sgRNA1 to specifically induce *Fgfr3*$^{G369C}$ allelic loci mutations in *Fgfr3*$^{G369C}$ sperm from *Fgfr3*$^{G369C}$ male mice, and we were interested in whether this could block dwarfism after the corrected embryos were transferred into the surrogate mothers, developed and were delivered as live pups. Remarkably, in the appearance between three groups of *Fgfr3*$^{G369C/+}$, *Fgfr3*$^{G369C\ del/+}$, and wild-type mice, *Fgfr3*$^{G369C/+}$ mice exhibit dome-shaped heads, shortened limb bones, and tails. However, compared with the *Fgfr3*$^{G369C/+}$ littermates, the *Fgfr3*$^{G369C\ del/+}$ mice all exhibited a normal body size, including heads and limbs similar to those of their wild-type littermates (Fig. 4c, d). From a radiological perspective, the rescued phenotype could also be supported by microcomputed tomography (Micro-CT) analysis as detailed below. The whole skeleton of the *Fgfr3*$^{G369C\ del/+}$ mouse was normal (Fig. 4e). The skull of the *Fgfr3*$^{G369C\ del/+}$ mouse was significantly returned to normal size compared with the mutant skull, and the significantly shortened femur and tibia were returned to the same level with wild-type mouse (Fig. 4f, g, Supplementary Fig. 8c). In addition, other deformed bone structure of the *Fgfr3*$^{G369C/+}$ mouse such as the abnormal intervertebral foramen, shortened jaw, and incisors were all significantly recovered in the *Fgfr3*$^{G369C\ del/+}$ mouse (Fig. 4f, Supplementary Fig. 8d, e). From the bone components analysis (see Methods for details), we performed general quantitative analysis of trabecular of the whole femurs between the three types of mice. Morphometric parameters of bones including trabecular bone volume fraction (BV/TV), trabecular number (TB.N), trabecular separation (Tb.Sp), and bone mineral density (BMD) revealed that BV/TV ($p < 0.0001$ vs wild type, df (2, 15), $F = 28.42$, one-way ANOVA) and TB.N ($p < 0.0001$ vs wild type, df (2, 15), $F = 417.0$) of all types of bones in the *Fgfr3*$^{G369C/+}$ mouse were significantly decreased, and Tb.Sp of them were significantly increased ($p < 0.0001$ vs wild type, df (2, 15), $F = 2072$). The change

tendencies of these parameters were all consistent with previous reports[31]. However, no or smaller difference in these parameters was seen between the wild-type mouse and the *Fgfr3*$^{G369C\ del/+}$ mouse ($p = 0.998$, df (2, 15), $F = 28.42$ in BV/TV, $p = 0.8859$, df (2, 15), $F = 417.0$ in TB/N, $p = 0.8617$, df (2, 15), $F = 2072$ in TB. Sp (each vs wild type), one-way ANOVA, Fig. 4h). Histology analysis also indicated that sparse bone trabecula in the primary and secondary ossification centers of *Fgfr3*$^{G369C/+}$ mice could be reversed in the *Fgfr3*$^{G369C\ del/+}$ mice (Fig. 4i). The combined analysis revealed that the bone phenotype of the *Fgfr3*$^{G369C/+}$ mice could be corrected by selectively disrupting the dominant allelic loci. Theoretically, Past-CRISPR strategies have extensive application for disrupting dominant pathogenic variants, suggesting a possible reference for silencing allelic loci of human disease genes.

## Discussion

By using traditional strategies, Cas9 targeting activity could endure into the two-cell or later stages, which could lead to the additional undirected genomic editing during embryonic development, which may be the reason for the high rates of generating genotypic mosaicism. In this study, we established a modified strategy that can efficiently restrict Cas9 targeting activity to a narrow temporal window before the first round of DNA replication in zygotes through pronuclear transfer-mediated dilution of Cas9 activity. Using this principle, we developed a gene editing method, Past-CRISPR, to specifically and efficiently target a specified parental genomic region, which dramatically reduces mosaicism (Fig. 5). This method can be applied to the rapid screening of imprinted and lethal gene functions in vivo and can be used in the rapid generation of animals with a unique genotype for therapeutic applications and disease model construction.

Numerous novel approaches based on the CRISPR system have been developed to expand the application areas even beyond gene editing functionality[33–35], but the CRISPR-Cas9 system remains the most widely used and efficient gene-editing technology in medicine and life sciences due to its high-efficiency DSB-mediated targeting and simplicity of operation. However, a great deal of work remains in enhancing the flexibility and precision of Cas9-induced gene. CRISPR-Cas9 can disrupt a gene by inducing DSBs to generate undesired editing byproducts (distinct indels) at the target sites[29,36]. Past-CRISPR utilizes the ability of Cas9 to induce DSBs to generate indels, but it only does so during the window before the PN4 zygote stage when Cas9 protein and sgRNA were exposed to only two sets of genomes. In this case, the

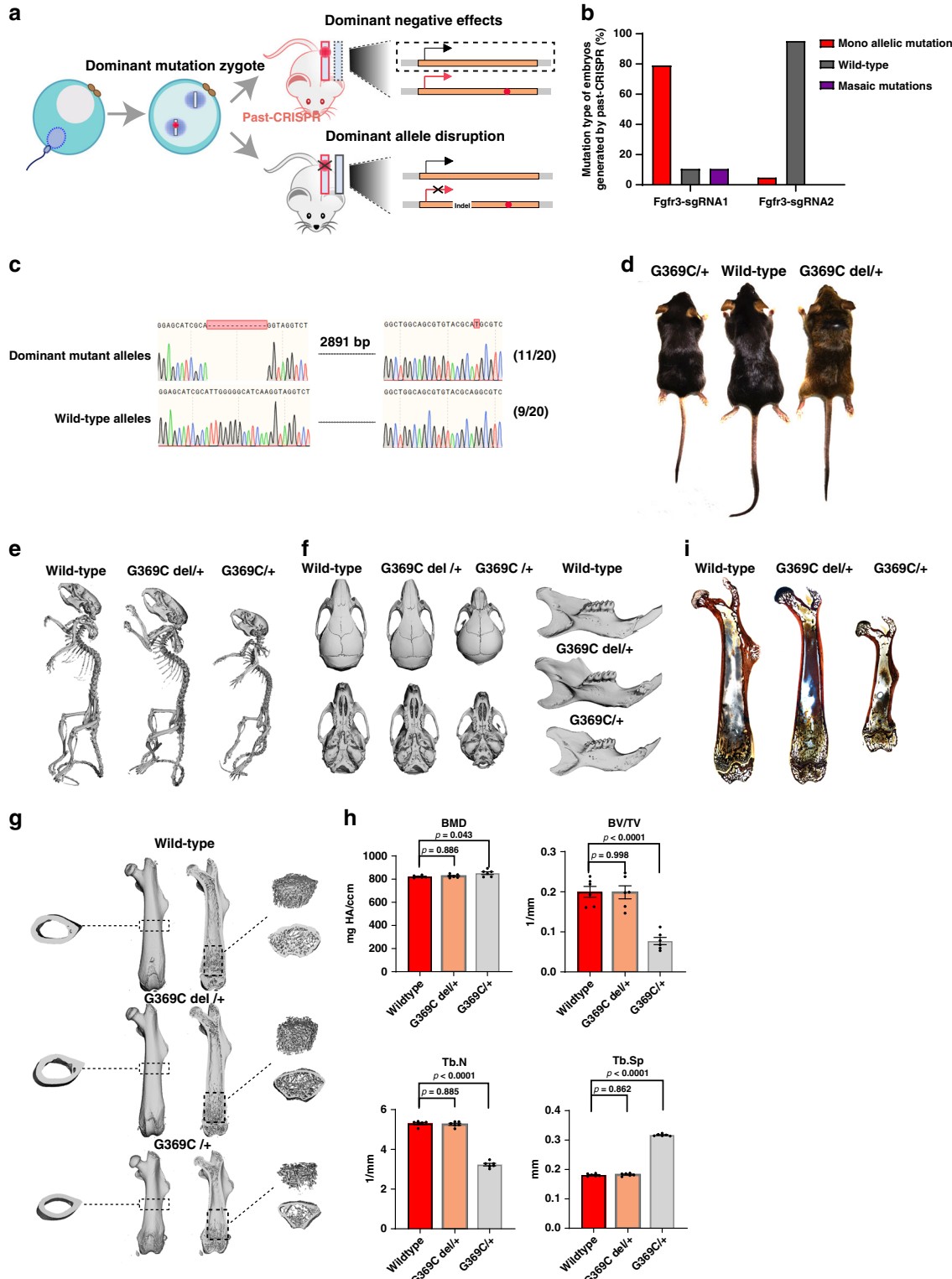

bi- or monoallelic gene mutations generated by Past-CRISPR could dramatically result in only one type of indel (identical indels), which largely avoided genotypic mosaic and reduced side effects produced by induced DSBs.

Meanwhile, because Cas9 activity is restricted to the period before the PN4 stage, the cells are in the early and mid S-phases of the cell cycle, and at this period (homology-directed repair) HDR mechanisms have actually been activated in the zygote[37–39]. Thus, it may be possible to achieve allele-specific knock-in based

on Past-CRISPR. However, recently, Anzalone et al. reported prime editing[40], a novel gene-editing strategy that could be applicable for postmitotic and primary cells and was independent of the cell cycle, and it achieved precise editing, providing us with another tool to incorporate when trying to achieve knock-in based editing.

The CRISPR-Cas9 system has been widely and efficiently used to generate gene knockout animals by zygote injection[7,41–43]. However, CRISPR-Cas9 could not selectively differentiate each of

**Fig. 4 Targeting strategies for specifically disrupting dominant alleles of Fgfr3**$^{G369C/+}$ **mice. a** Schematic of the dominant-negative effects exerted by dominant mutations and correction strategy by Past-CRISPR. Dominant mutations (red dot) do lead to the nonmutational impairment of the wild-type allele (black dashed line). Dominant mutation allele could be separated as the isolation of parental genome at the zygote pronuclear stage and disrupted by Past-CRISPR. **b** Percentage of induced mutation types in exon 4 of Fgfr3 by Past-CRISPR for each of two sgRNAs. The target genotype of each embryo was analyzed by TIDE. Two independent experiments were performed for every sgRNA. Source data are provided as a Source data profile. **c** TA cloning sequences resulting from the tail of one of the Fgfr3$^{G369C\ del/+}$ mice. The dominant mutation site is marked by the right red box. The indels induced by Fgfr3 sgRNA1 are marked by the left red box. A total of three Fgfr3$^{G369C\ del/+}$ mice were tested and 20 TA clones were sequenced in each sample. **d** Morphology of 2-month-old Fgfr3$^{G369C/+}$ mice (left), wild-type mice (middle), and Fgfr3$^{G369C\ del/+}$ mice (right). **e** Microcomputed tomography analyses show the skeletons of wild-type (WT), Fgfr3$^{G369C\ del/+}$, and Fgfr3$^{G369C/+}$ mice. **f** Microcomputed tomography analyses on the maxillofacial bone. Skulls and jaws are significantly reduced in size and mainly exhibit severe dysplasia in Fgfr3$^{G369C/+}$ mice compared with wild-type mice. The bone contours have been corrected to normal in Fgfr3$^{G369C\ del/+}$ mice. **g** Trabecular bone quantification analyses of femurs from wild-type (WT), Fgfr3$^{G369C\ del/+}$, and Fgfr3$^{G369C/+}$ mice. Femurs of Fgfr3$^{G369C/+}$ mice exhibit cortical bone thickness, noticeably sparse bone trabecular and shortened backbone. Femurs of Fgfr3$^{G369C\ del/+}$ mice are corrected to nearly normal levels. **h** Quantitative micro-CT analysis of distal femoral metaphysis from wild-type (WT), Fgfr3$^{G369C\ del/+}$, and Fgfr3$^{G369C/+}$ mice. Results are expressed as the mean ± SEM. of six technical replicates per group. P value (one-way ANOVA) are also shown. Source data are provided as a Source data profile. **i** Undecalcified bone section analysis of femurs shows primary sparse bone trabecula and secondary ossification centers of Fgfr3$^{G369C/+}$ mice and normal states in Fgfr3$^{G369C\ del/+}$ mice when compared to that of wild-type mice.

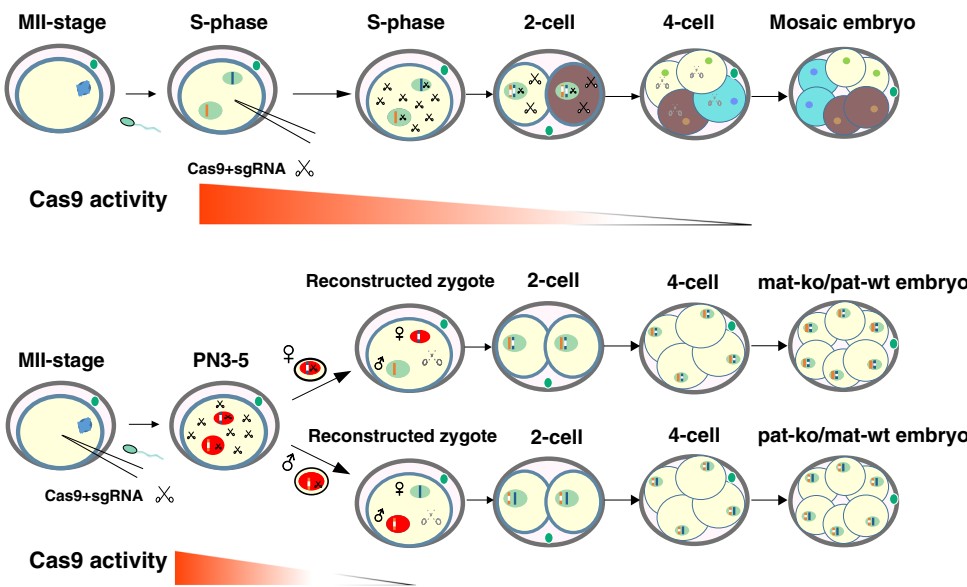

**Fig. 5 Model illustrating Cas9 activity during the development of embryos performed by traditional zygote injection methods in the upper panel and Past-CRISPR in the lower panel.** Solid scissors represent sufficient amounts of Cas9 with targeting activity, and dashed scissors represent low amounts of Cas9 with weak or no targeting activity. Different colors of different blastomeres represent different mutation genotypes.

the parental alleles, meaning that there is the same chance that each allele be edited by sgRNA targeting[18], which was verified by our studies where almost all of the embryos exhibited biallelic targeting. At present, a limited number of studies are exploring strategies to achieve allele-specific editing by the recognition of protospacer adjacent motifs (PAMs) or spacer region of sgRNAs for SNPs in the target DNA[44–48], but their usefulness has differed greatly due to inappropriate or sparse positions of PAMs or tolerations of mismatches that existed between the sgRNA and the target DNA by some Cas9 enzymes[49]. Allele-specific editing is limited by unstable allelic selectivity and a narrow[45] range of possible specific PAMs. However, Past-CRISPR could differentiate parental alleles and truly achieve allele-specific site editing with high efficiency and selectivity via isolating parental pronucleus and selectively targeting any sites from one or two of parental genomes at the zygotic pronuclear stage, which largely enhances the flexibility of CRISPR-Cas9 for genome editing.

Past-CRISPR could be widely used to improve various genome editing applications. As mentioned above, monoallelic knockout animal models carrying heritable lethal mutations could be generated efficiently in the F0 generation, while this is not achievable by traditional methods[18]. As several recent studies have revealed,

abundant parental-specific gene expression exerts a great influence on pre- or post-implantation development, but diseases resulting from these early changes are difficult to ascertain due to the unobservable symptom in early stages of pregnancy[24,50,51]. However, Past-CRISPR has great advantages in terms of the functional verification of individual parent-of-origin genes, including imprinted gene candidates. Meanwhile, mono- or biallelic gene knockout animal models with identical indels could be generated, which avoids undesired targeting outcomes to some extent. This can be used to generate haploinsufficient animal models for mimicking human disease[52] and complete gene knockout animal models for functional study. In addition, as a proof-of-principle, the ability to silence only mutant alleles, not wild-type alleles, could be used to correct the dominant gain-of-function mutations, such as the Fgfr3$^{G369C}$ mutations. More studies are needed to explore the wide range of function of these methods.

## Methods

**Experimental design.** Cas9 mRNA and individual sgRNA were together injected into MII oocytes. Wild-type MII oocytes and injected MII oocytes both were performed IVF at the same time. At 7–9 h post fertilization (h.p.f), most of

embryos have already developed into PN3–4 stage. We assessed that Cas9-induced DSBs can be resolved efficiently during this period[11]. Meanwhile, we supposed extended time of exposure to MII and zygote cytoplasm could allow CRISPR-Cas9 components to degrade partly before the first DNA replication of one-cell embryos. However, the DSB formation exactly falls within a window that can differentiate parental chromatins at the pronuclear stage (PN3–4) zygotes (Supplementary Fig. 1a). We could isolate the paternal and maternal pronuclei at the PN3–4 stage and exchange them into wild-type embryos that had been depleted paternal and maternal pronuclei, respectively. The cytoplasm of wild-type embryos could make residual Cas9 exists in edited pronulei further diluted and lose activity. Along with these reconstructed embryos developed to blastocysts, we then genotyped individual blastocyst each sample by PCR and TA clone sequencing of respective broken sites to identify which type of embryos they belong to (Fig. 1a). TA clone-sequencing outcomes from individual blastocyst typically exhibit genotypes of only a unique NHEJ mutation and wild type, suggesting that these Cas9-induced mutations at the parental mono-allele sites have been generated at the zygote stage of embryos and the activity of Cas9 is just limited before the first DNA replication. If the outcomes exhibit genotypes of significantly more distinct NHEJ mutations and wild type, suggesting it is more possible that the embryo is a heterozygote with mosaicism. If the outcomes of all clones exhibit NHEJ mutations, suggesting that the embryo is homozygous knockout (Supplementary Fig. 4b). If the outcomes of all clones exhibit wild type, suggesting that the embryo is wild type.

**Mouse embryo collection and embryo transplantation**. The specific patrogen-free mice were housed in the animal facility of Tongji University, Shanghai, China. The use and care of animals complied with the guideline of the Tongji University Guide for the Use of Laboratory Animals. MII oocytes were collected from 8-week-old BDF1 females and 8-week-old C57BL/6 females superovulated by injection with 5 IU each of pregnant mare serum gonadotropin (PMSG), followed by injection of 5 IU of human chorionic gonadotropin (hCG) (San-Sheng Pharmaceutical) 48 h later. Zygotes were obtained from the ampulla of the uterine tube of superovulated C57BL/6 or BDF1 female mice after mating with DBA or BDF1 male mice. CZB + Glutamin medium was used for culture of wild type and micromanipulative embryos at 37 °C under 5% $CO_2$ in air for 24 h, and 18–20 two-cell stage embryos were transferred into a oviduct of pseudopregnant ICR female at 0.5 d.p.c.

**Generation of Cas9 mRNA and sgRNA**. T7 promoter was added into Cas9 coding sequence by PCR amplification using PX330 vector and the primer T7-Cas9 (Supplementary Table 5). T7-Cas9 PCR products were collected through agarose gel electrophoresis and purified by QIAquick Gel Extraction Kit (Qiagen, USA), then used as the template for in vitro transcription (IVT) using mMESSAGE mMACHINE T7 ULTRA kit (Life Technologies). SgRNA template with T7 promotor at the 5′side by PCR amplification of px330. The T7-sgRNA PCR product was purified by QIAquick Gel Extraction Kit (Qiagen, USA) and used as the template for IVT using MEGA shortscript Transcription Kit (Life Technologies). Cas9 mRNA was purified by lithium chloride precipitation, and sgRNA was purified by phenol:chloroform extraction. sgRNA and mRNA aliquots were stored at −80 °C until use.

**MII oocyte and zygote injection**. MII oocytes were collected from superovulated eight-week-old C57 or BDF1 mice. Zygotes were collected from oviducts 24 h post hCG injection. For gene editing of embryos, the mixture of Cas9 mRNA or protein (100 ng $\mu l^{-1}$) and sgRNA (150 ng $\mu l^{-1}$) was injected into the cytoplasm of MII oocytes or zygotes by using a Piezo impact-driven micromanipulator in a droplet of HEPES-CZB medium.

**IVF of MII oocytes**. Female BDF1 or C57BL/6 mice (8 weeks old) were superovulated by injection with 5 IU of PMSG, followed by injection of 5 IU of hCG (San-Sheng Pharmaceutical) 48 h later. At 14 h after hCG injection, the cumulus oocyte complexes were released from the oviducts. MII oocytes were obtained from the ampulla of the uterine tube. Two cauda epidydimis were collected from one BDF1 or PWK male and cut several times with clippers. Then, they were transferred in a 1.5-ml eppendorf tube with 600 μl G-IVF medium under 5% $CO_2$ in air for 15–30 min to allow sperm to swim out through slits in the bases of cauda. At this time, spermatozoa capacitation was attained in the G-IVF medium. Activated sperm and oocytes were placed in the G-IVF droplets for 3–4 h, then washed and cultured in CZB or G1 medium.

**Pronuclear transfer**. MII oocytes were collected from 8-week-old superovulated BDF1 females and followed by IVF with BDF1 or PWK sperm. At 7–9 h.p.f, PN3–4 stage zygotes were collected based on the microscopic observation of the size of the two pronuclus and the distance between them, and transferred into HEPES-CZB medium containing 5 μg ml$^{-1}$ cytochalasin B and Nocodazole. They were then incubated for 5 min. Then, parental pronuclus were isolated from PN3–4 stage zygotes and incubated 4–5 s in the sendai virus (HVJ, Cosmo-bio), which was used for fusing pronuleus with cytoplasms, then injected into the perivetelline space of embryos that have been depleted another parental pronuclus by using a Piezo impact-driven micromanipulator. The isolated pronuclus were

fused into cytoplasms to reconstruct intact zygotes 0.5–1 h later. Then, the reconstructed zygotes were washed and cultured in CZB or G1 medium.

**Single-embryo PCR analysis**. Single embryos were picked up and washed for several times with HEPES-CZB medium by using a glass capillary, and then added into 5 μl G1 lysis solution (25 mM NaOH, 0.2 mM EDTA). Each tube was mixed thoroughly and centrifuged briefly. The G1 lysis was incubated at a temperature of 95 °C for 30 min and back to room temperature. Then 5 μl G2 lysis solution (40 mM Tris-HCl) was added to G1 lysis tubes. Each tube was mixed thoroughly and centrifuged briefly again. Then, the products of the lysis program were used as templates in PCR analysis. PCR amplification made use of ExTaq (Takara) enzyme. ExTaq (Takara) was denature at 94 °C for 3 min, and PCR was performed for 40 cycles. The anneal temperature and extended time were changed with different genes. PCR products were performed electrophoresis and extracted by using QIAquick Gel Extraction Kit (Qiagen, USA). Then Sanger sequencing was used to detect mutations.

**TIDE analysis**. About 500–1000 bp DNA fragments around the *Anapc2/Fgfr3*-targeting site were PCR amplified and sequenced from the single embryos that were treated with Cas9 mRNA and sgRNA. And the same DNA fragments were parallelly PCR amplified and sequenced from the control group. The sequencing outcomes of both groups and guide RNA sequences were inputted into the TIDE software[19] (https://tide.deskgen.com/). The parameters were set to the default maximus indel size of 10–15 nucleotides and the cover decomposition window was set to the largest possible window with high quality traces. The composite PCR amplified sequence trace can be decomposed into individual components by standard non-negative linear modeling of TIDE. The proportion of aberrant base signals could be visualized in an intuitive graph (Supplementary Fig. 2b). Meanwhile, the individual indel components of the aberrant sequences could be modeled and quantified at the cut site by the control sequence trace (Supplementary Fig. 2c). The *p* value of each indel is calculated by a two-tailed *t*-test of the variance–covariance matrix of the standard error.

**Single-cell PCR analysis**. We used a glass capillary under a dissection microscope to pick up embryos. 4-cell mouse embryos were digested with protease to remove the zona pellucida, and then the embryos were transferred into 0.5% BSA/PBS and gently pipetted to separate individual blastomere. Finally, the individual blastomere was transferred into a PCR tube containing cell lysis buffer (YIKONGenomics). Each tube was centrifuge to facilitate the mix. The procedure of cell lysis and amplification was followed the manufacturer's instructions. Then, the amplified products were used as templates in PCR analysis. PCR amplification made use of ExTaq (Takara) enzyme. ExTaq (Takara) was denature at 94 °C for 3 min, and PCR was performed for 34 cycles. The anneal temperature and extended time were changed with different genes. PCR products were performed electrophoresis and extracted by using QIAquick Gel Extraction Kit (Qiagen, USA).

**TA cloning and sequence analysis**. The genotypes of single blastocyst or mouse tissues were determined by PCR of genomic DNA extracted. The PCR products were purified and ligated to PLB-Simple vector (TIANGEN) and transformed to competent *Escherichia coli* strain DH5α. Culture at 37 °C overnight. Clones were picked to identify by PCR amplification. 2× Taq (TIANGEN) was denature at 95 °C for 3 min and PCR was performed for 34 cycles at 95 °C for 30 s, 58 °C for 30 s, and the extended time was changed with different genes. Positive clones were selected to identify by Sanger sequencing. The TA clone sequence outcomes from single blastocyst or cell were analyzed by Snap Gene software. The mutation genotype of single embryo was confirmed by sequence alignment with wild-type sequences.

**Immunofluorescent staining**. Fixation of freshly collected embryos from micromanipulated and cultured embryos in 4% paraformaldehyde for 30–60 min and permeabilized with 0.05% Triton-X-100 for 20 min. After permeabilization, embryos were washed 3× in PBS. And blocked for 3–4 h in blocking solution (3% BSA in PBS) and incubated with primary antibodies in 3% BSA (1:10). Antibody used were Anti-SpCas9 (Active Motif, Catalog No. 61758). After overnight incubation at 4 °C embryos were washed 3× in PBS, and incubated with second antibodies labeled with Alexa fluorophores (Invitrogen) in 3% BSA for 1–1.5 h at room temperature. After washing 3× in PBS, embryos were incubated with 4′-6-diamidino-2-phenylindole for visualizing DNA. The samples were observed under a confocal microscope LSM880.

**Microcomputed tomography and bone section analysis**. From the observation of appearance, the dwarfish-like features between all three *Fgfr3*[G369C del/+] mice generated by Past-CRISPR have been reversed. The body size of these mice have all returned to normal level similar to wild-type mice (Fig. 4d). One of the *Fgfr3*[G369C del/+] mice was picked at random for analyzing micro-CT. As a parallel control group, one *Fgfr3*[G369C/+] mouse and one wild-type mouse of the same age were also performed micro-CT analysis. For micro-CT analyses, the mice were perfused with 4% paraformaldehyde. Samples were dissected and preserved in 4% paraformaldehyde for microcomputed tomography (μCT 50 Scanco Medical). Skulls,

femurs, tibias, jaws, and vertebras were dismembered for some operation. Two-dimensional (2D) data were used from scanned slices (scanning thickness 10 μm), 3D analysis was used to calculate morphometric parameters of the whole femur bone, including BV/TV, trabecular thickness (Tb. Th), Tb.Sp and BMD. The morphometric parameters on the trabecular analysis were valuing by measuring six scattered cross-sections of the distal metaphysis (Fig. 4h and Supplementary Fig. 8f, g). For undecalcified bone sections analysis, bone tissues were fixed in 4% paraformaldehyde, dehydrated with gradient ethanol and embedded in photocured resin, and then cut into 50-μm sections. The different bone tissues were stained by Von kossa.

**ClinVar database analysis.** The ClinVar database was downloaded from NCBI[28] (accessed June 09.2020). The database was filtered for 'pathogenic' diseases, resulting in 81,137 entries. And the list of pathogenic variants was filtered for 'dominant' diseases, resulting in 1789 entries. We uploaded the resulting databases as Supplementary Data 1 and 2.

**Statistics and reproducibility.** For validation of methods, at least two independent experiments were performed for every tested gene, and the oocyte or embryo samples ($n = 30–50$) for each experiment were collected from more than one individual. Quantitative analysis (Figs. 3e, 4h and Supplementary Fig. 1b) was performed by GraphPad Prism 8.0. Error bars in the graphical data represent the standard error of mean (SEM). All statistical significance tests have been indicated in the corresponding figure legend. Statistical comparisons (Figs. 3e, 4h) were made using a one-way ANOVA with Tukey multiple hypothesis correction, $p$ values were considered significant at $*p < 0.05$, $**p < 0.01$, $***p < 0.001$ and $****p < 0.0001$.

**Reporting summary.** Further information on research design is available in the Nature Research Reporting Summary linked to this article.

## Data availability

All relevant data are reported in the main text or Supplementary Information. The source data underlying Figs. 3e, 4b, h, and Supplementary Figs. 1b, 6a, 7a are provided as Source Data files. Any additional data relevant to this manuscript are available from the authors upon reasonable request. Source data are provided with this paper.

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

## Acknowledgements

We thank Prof. Lin Chen for providing us the *Fgfr3*$^{G369C}$ mice. This work was primarily supported by the National Natural Science Foundation of China (31721003, 81630035, 31871448, 31871446). This work was also supported by the Ministry of Science and Technology of China (2017YFA0102602, 2016YFA0100400), the Shanghai Subject Chief Scientist Program (15XD1503500), the Shanghai excellent young medical personnel training program (2017YQ004), The key project of the Science and Technology of Shanghai Municipality (19JC1415300), the Shanghai municipal medical and health discipline construction projects (2017ZZ02015), the Shanghai Rising-Star Program (19QA1409600), the Young Elite Scientist Sponsorship Program by CAST (2018QNRC001) and the Fundamental Research Funds for the Central Universities (1515219049).

## Author contributions

W.L. and S.G. conceived and designed the experiments. Y.L. performed most of the experiments. Y.W., D.B., Y.J., Y.D.L., Y.Zhang, X.K., Y.Zhao, J.R., J.C., J.Y., H.W., X.T. and Z.W. assisted with the experiments. Y.L., W.L., and S.G. wrote the manuscript.

## Competing interests

The authors declare no competing interests.
