## [Peer Review File · Nature Communications]

Reviewers' Comments:

Reviewer #1:

Remarks to the Author:

This manuscript describes an interesting, and apparently fortuitously discovered, method for generating allele specific genome editing using a combination of CRISPR and pronuclear transfer. In a series of experiments, the authors demonstrate the validity of this approach and then a potential usage of this methodology, specifically for correcting dominant germline mutations. As far as I can tell, this is an entirely novel approach and the methodology is really interesting and has many potential applications. One generic criticism is that the authors do not specifically test the possible mechanism(s) by which this methodology works, but I do not think this should preclude publication. However, there are a few issues in terms of methods reporting and statistical analyses that do need addressing before publication. These are detailed below.

Although the number of embryos used in each experiment is stated, it is not clear from the Methods section or in the main text how many females were used to generate the embryos used in each experiment. For instance on Line 179 reads "Sequencing of 60 embryos revealed that 55 (91.7%) were mono allelic mutants..." Were all 60 embryos from one super-ovulated female? Or from several? This is quite important - the number should be stated, and I would argue that if <2 females were used, then additional data needs to be added to the experiments. This may be particularly relevant to the experiment described in Line 128 where only 9 embryos were used. The reason for this relates to the generality of the findings and what constitutes a biological replicate.

I am also concerned about the statistical tests used and the way they have been applied. In particular, I do not think that multiple separate t-tests are the most appropriate method that should be applied to the data presented in Figs 3b, 4h, and Fig S8f & g. Specifically, Figure 3 shows the analysis of two targeted imprinted genes, namely Mash2 and Peg3. From what I can gather, the authors have grouped both ko groups together (Mash2m-/p+ and Peg10m+/p-) and compared them to a wild-type group. Similarly they have grouped the opposite allele control groups together (Mash2m+/p- and Peg10m-/p+) and also compared them to the wild-type group also. There are two things wrong with this. Firstly, there is no reason in my mind to group the 'Mash2' animals with their equivalent 'Peg10' animals. These are effectively two separate tests of the ability to target an imprinted gene in an allele specific manner (which should be seen as a positive, but is lost by grouping). Secondly, for both the Mash2 and Peg10 experiments the authors have effectively two control groups - wild type and Mash2m+/p- or Peg10m-/p+ respectively. So the most appropriate way to statistically analyse these groups is to perform a one-way ANOVA with all three groups (e.g. Mash2m-/p+, Mash2m+/p- and wild-type) and then perform post-hoc analysis to assess the individual differences between the three groups. The issue of using ANOVA is also relevant for the data presented in Figure 4 (and Suppl. Figure 8). Again, here there are three groups - Ctrl, Blocked and point mutation. It is wrong to simply perform t-tests between Ctrl and point mutation, and then (separately) Ctrl and Blocked in order to test whether the point mutation group has a deficit that is rescued in the Blocked group. Instead, a one-way ANOVA should be conducted with all three groups included, and post-hoc test used to determine the difference between each of these groups. I must emphasise that I doubt very much that this will alter the interpretation of the data, which are quite clear. However, it is absolutely imperative that these data are analysed correctly in order to draw firm conclusions. It is also important that the authors report all the statistics correctly in the text - so for ANOVAs the F-value,

degrees of freedom and p-value should be reported.

Minor issues:

When representing parental specific allele manipulations it is usual to place the maternal allele first. So a Peg10 paternal knockout should be represented as "Peg10m+/p-" (or "Peg10M-wt/P-ko" if preferred)

Line 275 Do the authors really mean "rare animals"? I cannot envisage a situation where rare animals are likely to be genetically modified. Perhaps they mean "large and/of difficult to breed animals"?

Line 372-73 "while this not able to be achieved by traditional methods" is better phrased "while this is not achievable by traditional methods."

Line 89 of Methods - "ued" should be "used"

Reviewer #2:

Remarks to the Author:

In the present paper Li et al, present an innovative approach to optimise the use of the CRISPR-Cas9 system to create functional gene deletions in mice. CRISPR-Cas9 provides a way to create full-knockout mice in a single step, without the need of additional breedings, by introducing the Cas9 enzyme together with guide RNAs into the zygote. Whereas this approach is widely applied, the prolonged activity of the Cas9 enzyme creates multiple different knockout alleles as cell number increase during development. The resulting animal is thus a genetic mosaic which might confound the phenotype. Also this 'conventional' approach usually creates homozygous knockout animals and therefore does not allow easily for the creation of loss of function alleles with an embryonic lethal phenotype. Finally parental allele specific targeting is also not possible with current approaches.

Li et al., provide a solution whose elegance lies in its simplicity. Cas9 mRNA and guide RNAs are injected into an oocyte, which is then fertilized by IVT. After some incubation time the male and or the female pronucleus, containing the Cas9 modified DNA, is replaced with its counterpart in a wt zygote and allowed to develop in vitro or in vivo. The authors show nicely and convincingly that this approach allows the Cas9 enzyme to be efficiently diluted, which essentially removes genetic mosaicism, allows preparation of heterozygous deletions and controls the parental allele that is targeted. Nevertheless this approach needs a high level of skill in micro manipulating mouse embryos, which the authors seem to have mastered very well. Additionally this novel approach opens a new avenue for the use of Cas9 in the study of genes under the control of genomic imprinting. This is a highly important method that is relevant for a broad audience.

Whereas especially the growth and color phenotypes are very convincing and support the claims, the sequencing as well as the Fgfr3 rescue data need clarification at several points; and the general presentation of data and writing of text needs major improvements to be easily understood by the community, as detailed below.

General comments:

Standard gene name nomenclature should be correctly applied, for example 'Mash2' is written as 'mash2' at multiple occasions.

'mat swapped' and 'pat swapped' is shown in Figure 1 and mentioned in Figure 3 legend but not explained in the main text. As this likely refers to controls, the use of controls for each experiment should be clarified and clearly stated throughout the manuscript.

The wording for 'traditional zygote injection' and 'Past-CRISPR' should be streamlined. One example of confusing wording is the legend for Figure 2d and 2e.

The link between Supp. Table 2 and Sup. Figure S1d/e is unclear. Is there a link? If yes that should be clarified. If no - the purpose of these different pieces of data should be made clear.

Wherever bars with error bars are shown: Either the number of replicates should be stated in the Figure legend or in the figure itself, or the actual data points should be shown.

More specific comments:

Text:

Page 2 Line 25: 'accidentally learned observation': This is an interesting remark, but not explained in detail in the text. This should be either explained or removed

Page 4 Line 93/94: '... two differentiated parental chromatins ...' - It is unclear what the authors refer to here. Clearly the presence of two parental pronuclei in the zygote provide a unique window where the parental genomes are physically separated. If the authors refer to chromatin in the sense of DNA wrapped around histone proteins, this should be explained more clearly. Also the use of 'differentiated' is unclear in this context. If 'parental chromatins' is a commonly used expression, this should be referenced

Page 4 Line 98: A reference should be provided to support the claim that Anapc2 is necessary for pre implantation embryo development.

Page 4 Line 101: Here the authors state that pronuclei from PN3-4 were used – This seems to contrast with Figure 1a, where PN4-5 is indicated. The methods state that the pronuclei were isolated at the P4 stage. These differences should be clarified and/or streamlined.

Page 5 Line 125: 'TA clone sequence analysis' should be explained and the Methods referenced.

Page 5 Line 126: The wording "... into zygote-induced mutations ..." should be clarified.

Page 5 Line 138: "...consistent with previous studies." - a reference should be added.

Page 5 line 146: "The ratio between them was approximately equal." - is obvious from the numbers and should be removed

Page 5 Line 147: "TIDE analysis" should be briefly explained and the methods referenced. Also the meaning of the p-value analyzed by TIDE (Figure 4b, Sup. Fig. S2c) and of 'region for decomposition' Sup. Fig S2b should be explained to non-experts.

Pge 5 151: "... uniformly depleted of one Anapc2 allele ..." should be rephrased. Whereas the data directly shows that one allele was modified on the DNA level by Cas9, the mRNA of this allele was not investigated. It is possible that the mRNA was not depleted but rather does not produce functional protein any more.

Page 6 Line 159: 'Supp. Table 3' is referenced, but only contains gRNA target sites, here it would be useful to have a table similar to Supp. Table 2.

Page 7 Line 179: Presumably "... the 60 embryos ..." referred to in the text are 60/70 (85.7%) targeted embryos. This should be clarified and stated explicitly.

Page 7 Line 190ff: This paragraph explains Figure 2f, which essentially summarizes the whole manuscript. This feels out of place as it is in the middle of the manuscript - consider moving this to the end of the manuscript and describe in the discussion.

Page 8 Line 210/211: The sentence "... depletion of these genes caused an early embryonic lethal phenotype (Supplementary Fig. 5) ..." is unclear. Does this refer to a 'conventional' Cas9 gene deletion paradigm. This should be clearly stated.

Page 8 Line 217: "... previously published work." should be substantiated with References.

Page 8 Line 233 ff: The sentence starting with "Usually, none of the pups were born by injecting ..." should be rephrased. Were these genes tested in this study or is this a statement based on literature?

Page 9 Line 240: Referencing of Supp Table 4, which gives primer sequences, feels not entirely fitting here. Again a table similar to Sup Table 2 would be more appropriate at this position.

Page 9 Line 266/267: The meaning of the statement "... wild-type embryos or pups have a greater percentage of being manipulated by 2PN strategies ..." is unclear and should be clarified.

Page 8 Line 232 – Page 9 247: This paragraph describing the in vivo Anapc2 data, would fit much better with the rest of the Anapc2 data from Figure 1 and 2. Consider combining all Anapc2 data, which would it make easier for the reader to follow.

Page 10 Line 280-285: The point of this analysis at this point in the manuscript is unclear. Currently it appears that this is a discussion point to stress that many dominant mutant alleles exist in human disease, which could be corrected using this new technology.

Page 10 Line 293: Fgf3-dom should be called Fgf3-G369C.

Page 12 Line 342: The use of 'chromatin' in this context should be clarified.

Page 18: Line 586: 'Number, total embryos counted.' - Clarify to which number this statement refers to.

Page 19 Line 610: '... many independent experiments.' - here a concrete number should be given.

Figures:

Figure 1a: This figure would benefit from a legend explaining colors and symbols. Also a clear labeling of modified and wild-type pronuclei would help the understanding of this nice technology.

It appears that labeling for Figure 1b and 1c need to be swapped.

Figure 1c: Overview images should be presented that show the zygote/2-cell stage and surrounding space. It appears that the antibody staining of the zygote stage suffers from extensive background signal, which seems to be less severe at the 2-cell stage. Is this cytoplasmic staining? Also the white circle should be defined – is this the whole zygote or just one of the parental pronuclei?

Figure 1e: Providing actual sequencing results is informative but the labeling should be improved. Especially genotypes should be clearly stated instead of just using a color code.

Figure 2a: The indicated wild-type sequence should show both G and T nucleotides, why is this not the case?

Figure 2c/ Supp. Fig. S3d: Whereas the overall message seems somewhat clear the exact type of data that is shown is unclear. Please clearly indicate in the figure and in the legend which pronucleus was injected and which allele was analyzed.

Figure 3e: This panel would benefit from clear labeling of what is shown.

Figure 4a: The concept of functionally deleting a dominant mutant allele is not straight forward and should be explained by using a more schematic diagram. This would be much more helpful for the understanding of this concept than the very detailed explanation of guide RNA position (this could go to the Supplement).

Figure 4 general: The writing of 'blocked' should be streamlined. However 'blocked' seems to not accurately defining the genotype of these animals. The deletion introduced by the Cas9 activity in exon4 of the Fgfr3-G369C allele functionally deletes the dominant gene function. Should this allele not rather be called 'Fgfr3-G369C del'. This would be more informative than 'blocked'.

Also the number of biological replicates analyzed for each panel should be given, especially for 4c, and 4h.

Figure 4h: 'Ctrl' is written in the figure but in the legend (and the rest of the manuscript) wild-type is used – please streamline.

What is the difference between Figures 4h, S8f and S8g?. Please clearly state the difference between these graphs in the text.

Figure S1e would benefit from a legend.

Please give units to y-axis of Figure S8f and g

Figure S6c: The Legend indicates sequences of 5/6 alleles were analyzed. In the actual figure it appears that 6-8 alleles were analyzed.

Reviewer #3:

Remarks to the Author:

Li and colleagues attempt to control CRISPR-Cas9 activity in the generation of mutant mice using pronuclear transfer. A focus is on creating monoallelic, nonmosaic mutations in fertilized eggs so that mutation analysis is simpler (nonmosaic) and non-lethal for genes that may be required during embryogenesis (monoallelic). The approach is to co-inject Cas9 mRNA and sgRNA at fertilization of MII eggs, and then to take out the maternal or paternal pronucleus at the PN4-5 stage and place it into another fertilized egg in which the maternal or paternal pronucleus was removed, respectively. Another approach is to take out both pronuclei and place it into another fertilized egg in which both pronuclei have been removed, thereby preventing continued cleavage by Cas9, which remain in the initial egg.

The authors provide a large amount of evidence that the monoallelic, nonmosaic approach is feasible, which could be valuable in some circumstances. However, this approach requires a high degree of technical sophistication. It also requires twice the usual collection of mouse embryos. While mouse technicians in some core facilities may be able develop the skills – in particular the precise identification of the maternal and paternal pronuclei in every embryos, as well as their successful manipulation pronuclei – the trend has been to move away from injection to electroporation of mouse embryos, thereby requiring less skilled technicians. It is difficult to imagine therefore that this approach would be widely applied. While it is true that monoallelic loss of function would be required for embryonic lethal genes, this can usually be obtained in a fraction of embryos using current approaches, either through chance monoallelic mutation or in frame deletion on one allele. In addition, quenching Cas9 activity though anti-CRISPRs would likely be easier.

There are other concerns about the manuscript. The comparison is typically made to injection of Cas9 mRNA and sgRNA into fertilized eggs at the zygote stage. As far as I am aware, mouse engineering

facilities now typically use RNPs, rather than Cas9 mRNA injection, which could potentially prolong cleavage activity and hence mosaicism. Importantly, the control should be injection of MII eggs at fertilization without pronucleus removal, which is already known to lead to reduced mosaicism (ref 1). Moreover, it has been reported that the paternal genome is more readily edited in MII injected eggs, which would result in monoallelic mutations (Suzuki T Sci Rep 2014). Thus, it may be possible to reach the same goals as the authors without the technically challenging and resource consuming approach of pronuclear transfer,

Overall, while the authors achieve a technical tour de force with parental-specific pronuclear transfer, this manuscript is poorly written and extremely difficult to follow at many points. It also assumes a specialized knowledge of embryogenesis that general readers would not have (e.g., the meaning of PN4-5; what TIDE is, etc).

Point to point response:

First of all, we are delighted to receive the invaluable comments on our manuscript entitled "Precise allele-specific genome editing by spatiotemporal control of CRISPR-Cas9 via pronuclear transfer". We appreciate all three experienced reviewers for their positive comments in general. We also would like to thank them for providing us some constructive suggestions to further improve the quality of our manuscript. Based on the editor and reviewers' suggestions, we have performed additional experiments and revised our manuscript accordingly. We hope the revision is clear enough to address the major points raised by the reviewers. The major changes of the manuscript were summarized as follows.

- As reviewers suggested, we have modified statistical methods "one-way ANOVA" to analyze the data presented in Fig.3b, 4h and Fig S8f&g. As the reviewer might expect, there is a slight change in our data, but the tendency indicated in the analysis results is in the better direction. We showed this data in our point-to-point response and provided a table to report all the analysis results.
- We also noticed that reviewers raised the questions that the general presentation of data and writing of text needs major improvements to be easily understood by the community. In the revised manuscript, we have further organized the manuscript, and we have given more exhaustive complementary explanation and added some related supplementary tables and figures for some points difficult to understand.

We also addressed each specific point raised by the reviewers, as detailed below. The reviewers' comments are in black, and our replies are in red.

Reviewer #1

Although the number of embryos used in each experiment is stated, it is not clear from the Methods section or in the main text how many females were used to generate the embryos used in each experiment. For instance on Line 179 reads "Sequencing of 60 embryos revealed that 55 (91.7%) were mono allelic mutants..." Were all 60 embryos from one super-ovulated female? Or from several? This is quite important - the number should be stated, and I would argue that if <2 females were used, then additional data needs to be added to the experiments. This may be particularly relevant to the experiment described in Line 128 where only 9 embryos were used. The reason for this relates to the generality of the findings and what constitutes a biological replicate.

Answer: We appreciate the reviewer for this question. Actually, we have performed experiments for every tested gene at least two times, and oocyte or embryo samples (n=30~50) for each experiment were collected from more than one individual. In some experiments, most of the manipulated oocytes or zygotes were analyzed for pre-implantation development of embryos or transferred to pseudopregnant female mice, and several embryos were randomly selected to perform PCR amplification and sequencing. We have stated this point in the related Figure legends and methods section according to the suggestion, and we also added a new supplementary table related to these statistics of manipulated embryos for this section (Supp. Table 4).

I am also concerned about the statistical tests used and the way they have been applied. In

particular, I do not think that multiple separate t-tests are the most appropriate method that should be applied to the data presented in Figs 3b, 4h, and Fig S8f & g. Specifically, Figure 3 shows the analysis of two targeted imprinted genes, namely Mash2 and Peg3. From what I can gather, the authors have grouped both ko groups together (Mash2m-/p+ and Peg10m+/p-) and compared them to a wild-type group. Similarly they have grouped the opposite allele control groups together (Mash2m+/p- and Peg10m-/p+) and also compared them to the wild-type group also. There are two things wrong with this. Firstly, there is no reason in my mind to group the 'Mash2' animals with their equivalent 'Peg10' animals. These are effectively two separate tests of the ability to target an imprinted gene in an allele specific manner (which should be seen as a positive, but is lost by grouping). Secondly, for both the Mash2 and Peg10 experiments the authors have effectively two control groups - wild type and Mash2m+/p- or Peg10m-/p+ respectively. So the most appropriate way statistically analyse these groups is to perform a one-way ANOVA with all three groups (e.g. Mash2m-/p+, Mash2m+/p- and wild-type) and then perform post-hoc analysis to assess the individual differences between the three groups. The issue of using ANOVA is also relevant for the data presented in Figure 4 (and Suppl. Figure 8). Again, here there are three groups - Ctrl, Blocked and point mutation. It is wrong to simply perform t-tests between Ctrl and point mutation, and then (separately) Ctrl and Blocked in order to test whether the point mutation group has a deficit that is rescued in the Blocked group. Instead, a one-way ANOVA should be conducted with all three groups included, and post-hoc test used to determine the difference between each of these groups. I must emphasise that I doubt very much that this will alter the interpretation of the data, which are quite clear. However, it is absolutely imperative that these data are analysed correctly in order to draw firm conclusions. It is also important that the authors report all the statistics correctly in the text - so for ANOVAs the F-value, degrees of freedom and p-value should be reported.

Answer: We appreciate the enthusiasm, thorough review and helpful comments from this reviewer. As suggested by this reviewer, we realized that we used the inappropriate statistical tests for our data presented in Figs 3b, 4h, and Fig S8f & g. Therefore, we have performed the one-way ANOVA analysis for these groups according to the suggestions and reported all the statistics including the F-value, degrees of freedom and p-value in a new table. We are glad to see that the p-value results obtained from the new statistical analysis are consistent with we expected.

Minor issues:

1. When representing parental specific allele manipulations it is usual to place the maternal allele first. So a Peg10 paternal knockout should be represented as "Peg10m+/p-" (or "Peg10M-wt/P-ko" if preferred)

Answer: Thanks for the suggestion. All these points have been either revised or re-written in the revised manuscript.

2. Line 275 Do the authors really mean "rare animals"? I cannot envisage a situation where rare animals are likely to be genetically modified. Perhaps they mean "large and/of difficult to breed animals"?

Answer: We appreciate for the suggestion. It's our inappropriate description. And we have

changed the “rare” into “of difficult to breed animals”.

3. Line 372-73 "while this not able to be achieved by traditional methods" is better phrased "while this is not achievable by traditional methods."

4. Line 89 of Methods - "ued" should be "used"

Answer: We thanks for the comments on language editing. We have changed it into exact expression.

Reviewer #2

General comments:

1. Standard gene name nomenclature should be correctly applied, for example ‘Mash2’ is written as ‘mash2’ at multiple occasions.

Answer: We appreciate for the suggestion, and we have corrected all these points in the revised manuscript.

2. ‘mat swapped’ and ‘pat swapped’ is shown in Figure 1 and mentioned in Figure 3 legend but not explained in the main text. As this likely refers to controls, the use of controls for each experiment should be clarified and clearly stated throughout the manuscript.

Answer: Thanks for the suggestions. We are sorry that we did not describe it clearly in the previous version of the manuscript. We have given an interpretation for the “mat-swapped” and “pat-swapped” in the main text of the revised manuscript.

3. The wording for ‘traditional zygote injection’ and ‘Past-CRISPR’ should be streamlined. One example of confusing wording is the legend for Figure 2d and 2e.

Answer: We have corrected these mistakes in the legend of Figure 2d and 2e. And we have checked over the manuscript carefully, all related points have been revised or re-written in the revised manuscript.

4. The link between Supp. Table 2 and Sup. Figure S1d/e is unclear. Is there a link? If yes that should be clarified. If no - the purpose of these different pieces of data should be made clear.

Answer: Yes, it is. Actually, the sequencing outcomes of Supplementary Table 2 are the details of the Supplementary Figure S1e/f, and the Supplementary Table 1 are the details of the Supplementary Figure S1d. The sequences in the Supplementary Table 2 were mistakenly flipped into 3’-5’. We have corrected the direction of sequences in the Supplementary Table 2 and added an annotation into the Figure S1f legends.

5. Wherever bars with error bars are shown: Either the number of replicates should be stated in the Figure legend or in the figure itself, or the actual data points should be shown.

Answer: Thanks for the suggestions. We are sorry that we did not describe them clearly in some places of the previous manuscript. We went through the manuscript carefully, and found some missing details. All related points have been revised in the Figure legends or in the figures of the revised manuscript.

More specific comments:

Text:

1. Page 2 Line 25: ‘accidentally learned observation’: This is an interesting remark, but not explained in detail in the text. This should be either explained or removed

Answer: Thanks for pointing this out, it’s really an inappropriate description. We actually observed the phenomenon of the cytoplasmic dilution of Cas9 by immunostaining analysis and TA cloning analysis. Thus, we have replaced “accidentally learned observation” with “detailed observation”.

2. Page 4 Line 93/94: ‘... two differentiated parental chromatins ...’ - It is unclear what the authors refer to here. Clearly the presence of two parental pronuclei in the zygote provide a unique window where the parental genomes are physically separated. If the authors refer to chromatin in the sense of DNA wrapped around histone proteins, this should be explained more clearly. Also the use of ‘differentiated’ is unclear in this context. If ‘parental chromatins’ is a commonly used expression, this should be referenced

Answer: Thanks for the suggestion. We have corrected the meaning of this sentence in the revised manuscript.

3. Page 4 Line 98: A reference should be provided to support the claim that Anapc2 is necessary for pre implantation embryo development.

Answer: Thanks for the suggestion. We have added the reference in the revised manuscript.

4. Page 4 Line 101: Here the authors state that pronuclei from PN3-4 were used – This seems to contrast with Figure 1a, where PN4-5 is indicated. The methods state that the pronuclei were isolated at the P4 stage. These differences should be clarified and/or streamlined.

Answer: Sorry, it’s an inappropriate description. Actually, it is PN3-4 stage. We have revised these points in the revised manuscript.

5. Page 5 Line 125: ‘TA clone sequence analysis’ should be explained and the Methods referenced.

Answer: Thanks for the suggestion. We have added the details about the 'TA clone sequence analysis' in the Materials and Methods section and added an annotation in this place of the main text.

6. Page 5 Line 126: The wording "... into zygote-induced mutations ..." should be clarified.

Answer: We really appreciated the comments on language editing. We have corrected this grammatical error in the revised manuscript.

7. Page 5 Line 138: "...consistent with previous studies." - a reference should be added.

Answer: Yes, it is. We have inserted a related citation in the revised manuscript.

8. Page 5 line 146: "The ratio between them was approximately equal." - is obvious from the numbers and should be removed

Answer: Yes, it is. We have removed this sentence in the revised manuscript.

9. Page 5 Line 147: "TIDE analysis" should be briefly explained and the methods referenced. Also the meaning of the p-value analyzed by TIDE (Figure 4b, Sup. Fig. S2c) and of 'region for decomposition' Sup. Fig S2b should be explained to non-experts.

Answer: Thanks for the suggestion. We have given a more detailed description about TIDE analysis including the principle of decomposition and p-value in the methods section, and have added the commit into related Figure legend. We have added a briefly interpretation about the introduction and purpose of TIDE analysis.

10. Page 5 151: "... uniformly depleted of one Anapc2 allele ..." should be rephrased. Whereas the data directly shows that one allele was modified on the DNA level by Cas9, the mRNA of this allele was not investigated. It is possible that the mRNA was not depleted but rather does not produce functional protein any more.

Answer: Yes, it should be. On consideration, we think it's more appropriate to change "depleted" into "modified".

11. Page 6 Line 159: 'Supp. Table 3' is referenced, but only contains gRNA target sites, here it would be useful to have a table similar to Supp. Table 2.

Answer: We are sorry that we left it out in the previous manuscript. And we have added this related table into the supplementary table in the revised manuscript.

12. Page 7 Line 179: Presumably "... the 60 embryos ..." referred to in the text are 60/70 (85.7%) targeted embryos. This should be clarified and stated explicitly.

Answer: We are sorry that we did not describe this point clearly. We have stated this point more explicitly in this revised manuscript.

13. Page 7 Line 190ff: This paragraph explains Figure 2f, which essentially summarizes the whole manuscript. This feels out of place as it is in the middle of the manuscript - consider moving this to the end of the manuscript and describe in the discussion.

Answer: Thanks for the suggestion. We have moved this paragraph to the discussion part.

14. Page 8 Line 210/211: The sentence “ ... depletion of these genes caused an early embryonic lethal phenotype (Supplementary Fig. 5) ... “ is unclear. Does this refer to a ‘conventional’ Cas9 gene deletion paradigm. This should be clearly stated.

Answer: We are sorry that we did not describe it clearly in this place. Actually, these two imprinting genes reported previously were knocked out by the methods of homologous recombination. We verified that the gene deletion by Cas9 caused a lethal phenotype which is consistent with previous reports. We have stated this point more clearly in this revised manuscript.

15. Page 8 Line 217: “ ... previously published work.” should be substantiated with References.

Answer: Thanks for the suggestion. We have added references in the revised manuscript.

16. Page 8 Line 233 ff: The sentence starting with “Usually, none of the pups were born by injecting ...” should be rephrased. Were these genes tested in this study or is this a statement based on literature?

Answer: Yes, it's a statement based on literature. It's our inappropriate description. We have changed “these genes” into “embryonically lethal genes” and added related references to this place in the revised manuscript.

17. Page 9 Line 240: Referencing of Supp Table 4, which gives primer sequences, feels not entirely fitting here. Again a table similar to Sup Table 2 would be more appropriate at this position.

Answer: Yes, it is. We have made and placed a new related table here in the revised manuscript.

18. Page 9 Line 266/267: The meaning of the statement “... wild-type embryos or pups have a greater percentage of being manipulated by 2PN strategies ...” is unclear and should be clarified.

Answer: We are sorry that we did not describe this point clearly. The meaning of this statement is based on the Fig.3h, i. We have added an annotation at the end of this sentence in the revised manuscript.

19. Page 8 Line 232 – Page 9 247: This paragraph describing the in vivo Anapc2 data, would fit

much better with the rest of the Anapc2 data from Figure 1 and 2. Consider combining all Anapc2 data, which would make it easier for the reader to follow.

Answer: Thanks for the suggestion. According to the suggestion, in the revised manuscript we have changed the place of this section, and we have also made a new table (Supp. Table 4) include the in vitro verification experiments data of Anapc2, Peg10, Mash2 and the Anapc2, Peg10, Mash2 in vivo data. It would be easier for reader to follow.

20. Page 10 Line 280-285: The point of this analysis at this point in the manuscript is unclear. Currently it appears that this is a discussion point to stress that many dominant mutant alleles exist in human disease, which could be corrected using this new technology.

Answer: Yes, it's exactly what we want to express. We are interested in knowing how many dominant variants in human disease, which is also good for proposing the applicability of our methods.

21. Page 10 Line 293: Fgf3-dom should be called Fgf3-G369C.

Answer: Yes, it's should be. We have corrected it in the revised manuscript.

22. Page 12 Line 342: The use of 'chromatin' in this context should be clarified.

Answer: We have changed "chromatin" into "genome" in this revised manuscript.

23. Page 18: Line 586: 'Number, total embryos counted.' - Clarify to which number this statement refers to.

Answer: We are sorry that we did not describe it clearly in this place and have added the number in the revised manuscript.

24. Page 19 Line 610: '... many independent experiments.' - here a concrete number should be given.

Answer: Yes, we have corrected it in the revised manuscript.

Figures:

1. Figure 1a: This figure would benefit from a legend explaining colors and symbols. Also a clear labeling of modified and wild-type pronuclei would help the understanding of this nice technology.

It appears that labeling for Figure 1b and 1c need to be swapped.

Answer: Thanks for the suggestion. We have made some adjustments about Fig 1a and its legend in the revised manuscript.

2. Figure 1c: Overview images should be presented that show the zygote/2-cell stage and surrounding space. It appears that the antibody staining of the zygote stage suffers from extensive background signal, which seems to be less severe at the 2-cell stage. Is this cytoplasmic staining? Also the white circle should be defined – is this the whole zygote or just one of the parental pronuclei?

Answer: Thanks for the question. What we showed in this figure are one of the pronuclei at the zygote stage and one of the cell nuclei at the 2-cell stage. White lines demarcate the nuclear membrane. Actually, our purpose is to observe the Cas9 signal of nuclear region, because Cas9 enzyme is just able to take effects in the nuclear region. Thus, we show the images of nuclear regions of embryos, and we have given the explanations for it in the text and figure legend.

3. Figure 1e: Providing actual sequencing results is informative but the labeling should be improved. Especially genotypes should be clearly stated instead of just using a color code.

Answer: Thanks for your suggestion, we have added more details in the Figure 1e.

4. Figure 2a: The indicated wild-type sequence should show both G and T nucleotides, why is this not the case?

Answer: Yes, it should be. We have corrected it in the revised manuscript.

5. Figure 2c/ Supp. Fig. S3d: Whereas the overall message seems somewhat clear the exact type of data that is shown is unclear. Please clearly indicate in the figure and in the legend which pronucleus was injected and which allele was analyzed.

Answer: Thanks for the suggestion. We had remodified the Figure 2c/ Supp. Fig. S3d and added more clear statement in the related legend.

6. Figure 3e: This panel would benefit from clear labeling of what is shown.

Answer: Yes, it is. We had remodified this figure in the revised manuscript.

7. Figure 4a: The concept of functionally deleting a dominant mutant allele is not straight forward and should be explained by using a more schematic diagram. This would be much more helpful for the understanding of this concept than the very detailed explanation of guide RNA position (this could go to the Supplement).

Answer: Thanks for the suggestion. We have drawn a schematic diagram about this section and moved previous diagram into the Supp. Figure in the revised manuscript.

8. Figure 4 general: The writing of 'blocked' should be streamlined. However 'blocked' seems to not accurately defining the genotype of these animals. The deletion introduced by the Cas9

activity in exon4 of the Fgfr3-G369C allele functionally deletes the dominant gene function. Should this allele not rather be called 'Fgfr3-G369C del'. This would be more informative than 'blocked'.

Also the number of biological replicates analyzed for each panel should be given, especially for 4c, and 4h.

Also the number of biological replicates analyzed for each panel should be given, especially for 4c, and 4h.

Answer: Thanks for the suggestion. On consideration, we have changed "blocked" into "Fgfr3^{G369C del/+}". We have checked over the manuscript carefully, all related points have been revised or re-written in the revised manuscript.

We generated three mice in total by using our methods, and their genotypes were all G369C del/+ by TA cloning sequence analysis (Fig.4c). When they were 8 weeks old, by the observation of appearance, the three mice all exhibited normal body size including the normal heads and limbs similar to those of wildtype mice. Thus, we could easily identify that the bone phenotype of these three mice was all rescued (Fig.4d). The results of rescued phenotype had suggested that our methods could be applied to specifically deplete dominant mutant allele, which is the most important point we want to validate. As for Fig.4h, we randomly picked a Fgfr3^{G369C del/+} mouse from the three mice, one from Fgfr3^{G369C/+} mice and one wildtype mouse for bone components analysis. Our evaluation methods for it depend on measuring the different cross sections of distal metaphysis and performing six independent analyses per group. The accuracy of analysis results could be supported by the data of previous reports¹. We have shown more details in the text and methods section in the revised manuscript.

9. Figure 4h: 'Ctrl' is written in the figure but in the legend (and the rest of the manuscript) wild-type is used – please streamline.

Answer: We have corrected them in this revised manuscript.

10. What is the difference between Figures 4h, S8f and S8g?. Please clearly state the difference between these graphs in the text.

Answer: We are sorry that we did not describe this point clearly. We have rewritten this point in the text and stated the difference more clearly in this section of the revised manuscript.

11. Figure S1e would benefit from a legend.

Answer: Yes, it is. We have added more details in the Figure S1e legend.

12. Please give units to y-axis of Figure S8f and g

Answer: We have added units to y-axis in the revised manuscript.

13. Figure S6c: The Legend indicates sequences of 5/6 alleles were analyzed. In the actual figure

it appears that 6-8 alleles were analyzed.

Answer: It's an inappropriate description. Actually, it is 6-8 alleles. We have revised this point in the revised manuscript.

Reviewer #3

There are other concerns about the manuscript. The comparison is typically made to injection of Cas9 mRNA and sgRNA into fertilized eggs at the zygote stage. As far as I am aware, mouse engineering facilities now typically use RNPs, rather than Cas9 mRNA injection, which could potentially prolong cleavage activity and hence mosaicism.

Answer: Thanks for the suggestion. It was just as the reviewer described, previous studies also suggest that the direct expression of Cas9 protein in early-stage zygotes could reduce mosaic mutations. But we think the strategies of introduction of Cas9 protein/ sgRNA ribonucleoprotein (RNP) into zygotes also have some limitations. First, in order to decrease the mosaicism, the strategies have stricter requirement in the delivery time². To overcome this problem, it is necessary to introduce the Cas9 RNPs into very early-stage zygotes prepared by IVF. But introduction of Cas9 RNPs into zygotes prepared after natural breeding can not overcome the issue of mosaicism^{2, 3}. To validate it again, we performed the injection of Cas9 protein and sgRNA targeting Tyr gene into in vivo zygotes after natural breeding and genotyped 6 edited embryos. The sequencing results showed that all the embryos were mosaic mutants, similar to embryos edited by Cas9 mRNA and sgRNA (Table. a). Second, unlike Cas9 mRNA, Cas9 proteins may have an unstable targeting efficiency on different species⁴. For example, in non-human primates, Cas9 RNPs neither increase the targeting efficiency nor reduces the rate of mosaicism. Thus, we need to take into account many considerations including delivering stage, the quality of Cas9 proteins and species difference of embryos, etc.

a

strategies	Cas9/sgRNA	Total embryos	Number of mutations							Average mutations
			1	2	3	4	5	6	7	
Zygote injection	Cas9 protein+sgRNA (Tyr)	6	0	0	2	3	1	0	0	3.83
MII injection	Cas9 mRNA+sgRNA (Anapc2)	24	6	6	5	4	1	1	1	2.83
Zygote injection	Cas9 mRNA+sgRNA (Anapc2)	18	2	5	8	3	0	0	0	2.67

Importantly, the control should be injection of MII eggs at fertilization without pronucleus removal, which is already known to lead to reduced mosaicism (ref 1). Moreover, it has been reported that the paternal genome is more readily edited in MII injected eggs, which would result

in monoallelic mutations (Suzuki T Sci Rep 2014). Thus, it may be possible to reach the same goals as the authors without the technically challenging and resource consuming approach of pronuclear transfer,

Answer: Thanks for the suggestion. To validate it, we performed injection of 100ng/μl Cas9 mRNA and 150ng/μl sgRNA targeting *Anapc2* gene into MII oocytes, and then performed IVF about one hour later. We totally genotyped 24 edited embryos for the CRISPR/Cas9-induced mutations in the targeting *Anapc2* locus. However, we analyzed that by comparing the number of mutant alleles in every embryo between the MII injection group and zygote injection group (Table. a), there was no significant difference in the genome editing efficiency and the ratio of mosaicism (75% versus 88.9%) between the two groups (Fig. b). The sequencing results also showed that only one of 24 (~4.17%) embryos produced by MII oocyte injection strategy was a monoallelic mutant (Fig. c).

Suzuki T et al. reported an important point that paternal allelic editing was inclined to occur during the early stage of development and during this stage maternal genome was inherently refractory to editing⁵. But this strategy has not yet succeeded in achieving parental allele-specific editing. A recent study⁶ also reported that they performed gene editing by screening different ratios of Cas9 mRNA/sgRNA concentrations, but even if microinjecting the very low concentration of Cas9 mRNA/sgRNA, it still could not efficiently and consistently generate monoallelic mutants, which suggested that it was difficult to achieve precise allele-specific editing by strategies of controlling the injection concentrations.

Overall, while the authors achieve a technical tour de force with parental-specific pronuclear transfer, this manuscript is poorly written and extremely difficult to follow at many points. It also assumes a specialized knowledge of embryogenesis that general readers would not have (e.g., the meaning of PN4-5; what TIDE is, etc).

Answer: Thanks for the suggestion. We are sorry that we did not describe these related contents clearly in the previous manuscript. We have re-written and revised related points in the revised manuscript. We have given a more detailed description about TIDE analysis in the methods section, and have added the commit into related Figure legend. We have added a briefly interpretation about the introduction and purpose of TIDE analysis in the text. And we have given a schematic diagram to explain the developmental stage (PN 1-5) of zygotes, which should be

more helpful for the understanding.

References

1. Su, N. et al. Gain-of-function mutation in FGFR3 in mice leads to decreased bone mass by affecting both osteoblastogenesis and osteoclastogenesis. *Hum Mol Genet* **19**, 1199-1210 (2010).
2. Hashimoto, M., Yamashita, Y. & Takemoto, T. Electroporation of Cas9 protein/sgRNA into early pronuclear zygotes generates non-mosaic mutants in the mouse. *Dev Biol* **418**, 1-9 (2016).
3. Sung, Y.H. et al. Highly efficient gene knockout in mice and zebrafish with RNA-guided endonucleases. *Genome Res* **24**, 125-131 (2014).
4. Tu, Z. et al. Promoting Cas9 degradation reduces mosaic mutations in non-human primate embryos. *Sci Rep* **7**, 42081 (2017).
5. Suzuki, T., Asami, M. & Perry, A.C. Asymmetric parental genome engineering by Cas9 during mouse meiotic exit. *Sci Rep* **4**, 7621 (2014).
6. Yi Wu, J.Z., Boya Peng, Dan Tian, Dong Zhang, Yang Li, Xiaoyu Feng, Jinghao Liu, Jun Li, Teng Zhang, Xiaoyong Liu, Jing Lu, Baian Chen, Songlin Wang Generating viable mice with heritable embryonically lethal mutations using the CRISPR-Cas9 system in two-cell embryos. *Nature communications* **10.1 (2019): 1-13**. (2019).

Reviewers' Comments:

Reviewer #1:

Remarks to the Author:

In their response letter the authors indicate they have addressed all my previous criticisms. Whilst I do not doubt this, the statistics (F values, T-values, P-values etc) are still not reported in the main Results section. Instead the authors indicate they are in supplementary files, but which of the many supplementary files contain the relevant statistics is not clearly indicated in the text.

Moreover, in my view it is not acceptable to put statistics pertaining to main results in supplementary data files - these files should be used for the providing the actual raw data and/or for presenting additional data that supports the main findings. Please include any statistical analyses alongside results when and where they are reported in the main text.

Reviewer #2:

Remarks to the Author:

The authors adequately and sufficiently addressed my comments in their revision – except for one, see below:

Previous Point not adequately addressed:

I still do not find the connection between Fig. 4H and Fig. S8f,g. The authors replied that for Fig. 4H they “ ... randomly picked a Fgfr3 G369C del/+ mouse from the three mice, one from Fgfr3 G369C/+ mice and one wild-type mouse ...”. The method section indicates that only one mouse of each genotype was used and states: “The morphometric parameters of each types of bone on the trabecular analysis were valuing by measuring six scattered cross sections of distal metaphysis (Fig.4h, Supplementary Fig.8f, g)”. Does that mean that the 3 separate figures (Fig. 4H and Fig. S8f,g) are technical replicates? This should be stated clearly in the text.

Upon reading of the manuscript I found a number of minor points that still need attention:

Page 6 Line 156: Is the phrase 'modified of' correct?

Page 10, Line 293: space between references!

Page 20 Line 581: Sup.Table 5,6 are now 7,8

Page 11 Line 327: “bone phenotype of the Fgfr3 G369C del/+ mice” - do the authors mean: Fgfr3 G369C

Page 13 Line 382: Do the authors mean “improve” instead of “approve”?

Figure 3g,i normal/wt should be streamlined

Sup.Fig.1: Legend: um should be ym

Description of Tables S7, S8 should be up dated

Is Table S6 mentioned in the text?

Table S5 is not correctly referenced – should go with Fig. 3E

As a general comment the grammar especially of the newly added parts of the figure legends and use of words in the Methods should be double-checked.

Reviewer #3:

Remarks to the Author:

Cas9-gRNA RNPs are frequently used for zygote injections. As mentioned in the previous review, the control should be injection of MII eggs with Cas9 RNP at fertilization without pronucleus removal, which is already known to lead to reduced mosaicism.

Point to point response:

Reviewer #1 (Remarks to the Author):

In their response letter the authors indicate they have addressed all my previous criticisms. Whilst I do not doubt this, the statistics (F values, T-values, P-values etc) are still not reported in the main Results section. Instead the authors indicate they are in supplementary files, but which of the many supplementary files contain the relevant statistics is not clearly indicated in the text.

Moreover, in my view it is not acceptable to put statistics pertaining to main results in supplementary data files - these files should be used for the providing the actual raw data and/or for presenting additional data that supports the main findings. Please include any statistical analyses alongside results when and where they are reported in the main text.

Answer: We thank the reviewer for this suggestion. We are sorry that we did not describe it clearly in the previous revision. In this further revised manuscript, we have checked over the manuscript and put the statistics data to the relevant main results in the results section.

Reviewer #2 (Remarks to the Author):

The authors adequately and sufficiently addressed my comments in their revision – except for one, see below:

Previous Point not adequately addressed:

I still do not find the connection between Fig. 4H and Fig. S8f,g. The authors replied that for Fig. 4H they “ ... randomly picked a Fgfr3 G369C del/+ mouse from the three mice, one from Fgfr3 G369C/+ mice and one wild-type mouse ...”. The method section indicates that only one mouse of each genotype was used and states: “The morphometric parameters of each types of bone on the trabecular analysis were valuing by measuring six scattered cross sections of distal metaphysis (Fig.4h, Supplementary Fig.8f, g)”. Does that mean that the 3 separate figures (Fig. 4H and Fig. S8f,g) are technical replicates? This should be stated clearly in the text.

Answer: We thank the reviewer for this suggestion. Indeed, we did not state it clearly in the previous revision. The three separate figures (Fig.4h and Fig. S8f, g) are technical replicates. We have stated them clearly in the legend of the Fig.4h and the method part. To simplify the results, we deleted the Fig.S8f, g in the revised manuscript.

Upon reading of the manuscript I found a number of minor points that still need attention:

Page 6 Line 156: Is the phrase ‘modified of’ correct?

Page 10, Line 293: space between references!

Page 20 Line 581: Sup.Table 5,6 are now 7,8

Page 11 Line 327: “bone phenotype of the Fgfr3 G369C del/+ mice” - do the authors mean:

Fgfr3 G369C

Page 13 Line 382: Do the authors mean “improve” instead of “approve”?

Figure 3g,i normal/wt should be streamlined

Sup.Fig.1: Legend: um should be ym

Answer: We appreciate this suggestion, we have corrected the above points in the revised manuscripts.

Description of Tables S7, S8 should be up dated

Answer: We have downloaded the latest ClinVar database (June 09 2020) and updated the screening data in the revised manuscript.

Is Table S6 mentioned in the text?

Answer: We have mentioned it in the methods section of the revised manuscript.

Table S5 is not correctly referenced – should go with Fig. 3E

Answer: We appreciate this suggestion. We have moved Table S5 to the source data section in the revised manuscript.

As a general comment the grammar especially of the newly added parts of the figure legends and use of words in the Methods should be double-checked.

Answer: We appreciate this suggestion. We have checked over the manuscript and revised accordingly.

Reviewer #3 (Remarks to the Author):

Cas9-gRNA RNPs are frequently used for zygote injections. As mentioned in the previous review, the control should be injection of MII eggs with Cas9 RNP at fertilization without pronucleus removal, which is already known to lead to reduced mosaicism.

Answer: We thank the reviewer for this suggestion. We have added the Cas9 RNP injection in MII eggs experiments as a control group. To validate the efficiency of this strategy, we injected Cas9 protein and sgRNAs into MII oocytes and then performing IVF. We mainly tested 46 Anapc2 edited embryos and 27 Tyr edited embryos by Sanger sequencing. And we transferred Tyr edited embryos to pseudopregnant female mice and produced 20 pups.

We analyzed the efficiency of monoallelic targeting and mosaicism to the Anapc2 and Tyr edited embryos. The results showed that 45.7% (21/46) of the Anapc2 edited embryos and 44.44% (12/27) of Tyr edited embryos generated by Cas9 RNP injection in M-phase stage oocytes showed mosaicism, which was obviously lower than that of embryos (Anapc2 edited: 76.5%; Tyr edited:

81.8%) generated by zygote injection strategy, but still higher than that of embryos (average 8.3%) generated by Past-CRISPR strategy (Fig.1b, d). The efficiency of monoallelic targeting was also low. Only one of 42 *Anapc2* targeting embryos and one of 27 *Tyr* targeting embryos were monoallelic mutants (Fig. 1a, d).

Pigmentation phenotypes of the newborn mice from *Tyr* CRISPR/Cas9 RNP injection into MII oocyte showed that 30% (6/20) of them were mosaicism and 20% (4/20) of them were albino (Fig. 1c, e).

Fig.1a. Pie charts show the percentage of the three predicted targeting genotypes in the Past-CRISPR- edited embryos (upleft), traditional zygote injection embryos (upright) and Cas9 RNP injection in M-phase embryos (downleft). Numbers in pie charts represent the percentage of embryos. All the tested embryos were from at least six independent experiments.

Fig.1b. Pie charts showing percentages of different genotypes of all embryos treated with Past-CRISPR (upleft), traditional zygote injection methods (upright) and Cas9 RNP injection in M-phase (downleft). Numbers in pie charts represent the percentage of embryos. All the tested embryos were from at least three independent experiments.

Fig.1c. Percentages of different phenotypes of mice resulting from 2PN transfer, Cas9 RNP injection in M-phase and traditional zygote injection. The number above each column represents the total mice counted.

Fig.1d. Percentages of different mutation types of blastocysts resulting from 2PN-transfer, Cas9 RNP injection in M-phase and traditional zygote injection. The number above each column represents the total number of embryos analyzed.

Fig.1e. Pigmentation phenotypes of representative litters resulting from 2PN-transfer (left), Cas9 RNP injection in M-phase and traditional zygote injection (right). White arrowhead, albino; red arrowhead, wildtype; green arrowhead, spotted.

Reviewers' Comments:

Reviewer #2:

Remarks to the Author:

All points raised in the previous round of review were adequately addressed, data presentation of Figure 4 is much clearer now. Also quite a bit of new data was added in response to reviewer 3 (Crispr-RNP) which further stresses the improvements of the method. I have no further concerns.

Reviewer #3:

Remarks to the Author:

The addition of the RNP injection method for comparison is a valuable addition to the manuscript. I would suggest, however, labeling the pie charts in Fig. 2d,e with the method used for each.

REVIEWERS' COMMENTS:

Reviewer #2 (Remarks to the Author):

All points raised in the previous round of review were adequately addressed, data presentation of Figure 4 is much clearer now. Also quite a bit of new data was added in response to reviewer 3 (Crispr-RNP) which further stresses the improvements of the method. I have no further concerns.

Answer: We appreciate the reviewer's positive comments.

Reviewer #3 (Remarks to the Author):

The addition of the RNP injection method for comparison is a valuable addition to the manuscript. I would suggest, however, labeling the pie charts in Fig. 2d,e with the method used for each.

Answer: We thank the reviewer for this suggestion. We have labeled the pie charts in Fig.2d, e with the respective method used according to the reviewer's comment.